# Transient IGF-1R inhibition combined with osimertinib eradicates AXL-low expressing *EGFR* mutated lung cancer

Rong Wang [1,2], Tadaaki Yamada [1,3,14✉], Kenji Kita [1], Hirokazu Taniguchi [1,4], Sachiko Arai[1], Koji Fukuda[1,5], Minoru Terashima [6], Akihiko Ishimura [6], Akihiro Nishiyama[1], Azusa Tanimoto [1], Shinji Takeuchi [1,5], Koshiro Ohtsubo[1], Kaname Yamashita[1], Tomoyoshi Yamano [7], Akihiro Yoshimura [3], Koichi Takayama[3], Kyoichi Kaira [8], Yoshihiko Taniguchi [9], Shinji Atagi [9], Hisanori Uehara[10], Rikinari Hanayama [5,7], Isao Matsumoto [11], Xujun Han [1,5,12], Kunio Matsumoto [5,12,13], Wei Wang [2,14✉], Takeshi Suzuki[6,13] & Seiji Yano [1,5,14✉]

Drug tolerance is the basis for acquired resistance to epidermal growth factor receptor-tyrosine kinase inhibitors (EGFR-TKIs) including osimertinib, through mechanisms that still remain unclear. Here, we show that while AXL-low expressing *EGFR* mutated lung cancer (*EGFR*mut-LC) cells are more sensitive to osimertinib than AXL-high expressing *EGFR*mut-LC cells, a small population emerge osimertinib tolerance. The tolerance is mediated by the increased expression and phosphorylation of insulin-like growth factor-1 receptor (IGF-1R), caused by the induction of its transcription factor FOXA1. IGF-1R maintains association with EGFR and adaptor proteins, including Gab1 and IRS1, in the presence of osimertinib and restores the survival signal. In AXL-low-expressing *EGFR*mut-LC cell-derived xenograft and patient-derived xenograft models, transient IGF-1R inhibition combined with continuous osimertinib treatment could eradicate tumors and prevent regrowth even after the cessation of osimertinib. These results indicate that optimal inhibition of tolerant signals combined with osimertinib may dramatically improve the outcome of *EGFR*mut-LC.

[1] Division of Medical Oncology, Cancer Research Institute, Kanazawa University, Kanazawa, Japan. [2] Department of Radiation Oncology, Nanfang Hospital, Southern Medical University, Guangzhou, China. [3] Department of Pulmonary Medicine, Graduate School of Medical Science, Kyoto Prefectural University of Medicine, Kyoto, Japan. [4] Department of Respiratory Medicine, Nagasaki University Graduate School of Biomedical Sciences, Nagasaki, Japan. [5] Nano Life Science Institute, Kanazawa University, Kanazawa, Japan. [6] Division of Functional Genomics, Cancer Research Institute, Kanazawa University Kanazawa, Kanazawa, Japan. [7] Department of Immunology, Graduate School of Medicine, Kanazawa University, Kanazawa, Japan. [8] Department of Respiratory Medicine, Comprehensive Cancer Center, International Medical Center, Saitama Medical University, Hidaka, Japan. [9] Department of Thoracic Oncology, National Hospital Organization Kinki-chuo Chest Medical Center, Sakai, Japan. [10] Division of Pathology, Tokushima University Hospital, Tokushima, Japan. [11] Department of Thoracic, Cardiovascular and General Surgery, Kanazawa University, Kanazawa, Japan. [12] Division of Tumor Dynamics and Regulation, Cancer Research Institute, Kanazawa University, Kanazawa, Japan. [13] Tumor Microenvironment Research Unit, Institute for Frontier Science Initiative, Kanazawa University, Kanazawa, Japan. [14] These authors jointly supervised this work: Tadaaki Yamada, Wei Wang, Seiji Yano. ✉email: tayamada@koto.kpu-m.ac.jp; wangwei9500@hotmail.com; syano@staff.kanazawa-u.ac.jp

Lung cancer is the leading cause of cancer-related deaths worldwide, with non-small cell lung cancer (NSCLC) comprising 85% of lung cancers. Immune checkpoint inhibitors, including the anti-programmed death 1 (PD-1) antibody and the anti-programmed death-ligand 1 (PD-L1) antibody, have been approved for the treatment of advanced-stage NSCLC and have been shown to cure a population of NSCLC patients even at an advanced stage[1,2]. The therapeutic potential of ICIs encourages the development of novel drug therapies which would dramatically improve the prognosis of advanced NSCLC.

Osimertinib is a third-generation epidermal growth factor receptor-tyrosine kinase inhibitor (EGFR-TKI), which selectively inhibits mutated *EGFR* (e.g., *EGFR* with exon 19 deletion, L858R, and T790M) and spares wild type EGFR[3]. Osimertinib is effective in T790M-positive *EGFR*-mutated NSCLC patients refractory to classical type EGFR-TKIs, including gefitinib and erlotinib[4]. Furthermore, osimertinib is more efficient than gefitinib and erlotinib when given as the first-line treatment in *EGFR*-mutated NSCLC[5]. However, almost all the patients will experience disease recurrence due to acquired resistance to osimertinib. Regarding acquired resistance to osimertinib, several mechanisms including resistance mutations in *EGFR* (C797S/C796D), *MET* amplification, and the emergence of other driver oncogenes (*KRAS* mutations, *BRAF* mutations, *RET* fusion, etc.) have been reported in *EGFR*-mutated lung cancer patients with or without *EGFR*-T790M mutation[6–9]. Considering the fact that the *EGFR*-T790M mutation occurs in a large population (60–70%) of *EGFR*-mutated lung cancer patients who acquired resistance to first-generation EGFR-TKIs gefitinib and erlotinib[10], these observations suggest that mechanisms of osimertinib resistance are much more diverse, and thus acquired osimertinib resistance may be harder to be controlled. Recent studies have uncovered that a small population of cells adapts to the initial treatment with EGFR-TKIs as persisters, presenting the basis for acquired resistant lesions[11]. By elucidating the adaptation mechanism following the initial treatment with EGFR-TKIs, we could develop novel initiation therapies to eradicate tumor cells, and thereby further improve the outcome of advanced *EGFR*-mutated NSCLC by preventing the development of acquired resistance.

Previously, we reported that in AXL-high-expressing *EGFR*-mutated NSCLC, a small population of tumor cells emerged tolerant to osimertinib as persisters by restoring the survival signal from AXL associated with EGFR and HER3, and the combined treatment with osimertinib and an AXL inhibitor prevented the development of acquired resistance to osimertinib[12]. On the other hand, even in AXL-low-expressing *EGFR*-mutated NSCLC, a small population of tumor cells persist to osimertinib and develop acquired resistant tumors. However, the underlying mechanisms remain unknown.

In the present study, we investigated the mechanism of tolerance to osimertinib in AXL-low-expressing *EGFR*-mutated NSCLC cell-line-derived xenografts (CDX) and a patient-derived xenograft (PDX). We observed that the AXL-low-expressing cells demonstrated an increased level of phosphorylated insulin-like growth factor receptor 1 (IGF-1R) and emerged tolerant to osimertinib exposure by restoring the survival signal from IGF-1R associated with EGFR. The transient combination of the IGF-1R inhibitor with continuous osimertinib eradicated the tumor cells and prevented the regrowth in CDX and PDX models of AXL-low-expressing *EGFR*-mutated NSCLC.

## Results

**Osimertinib increased phospho-IGF-1R in AXL-low tumor cells.** In the first set of experiments, we compared the susceptibility to osimertinib in AXL-high-expressing (PC-9, PC-9/GXR,

and HCC4011) and AXL-low- expressing (HCC4006, HCC827, and H3255) *EGFR*-mutated NSCLC cell lines in vitro (Fig. 1 a, b). While osimertinib inhibited the viability of all *EGFR*-mutated NSCLC cell lines tested in a dose-dependent manner, the IC50 (half-maximum inhibitory concentration) values were lower in AXL-low-expressing tumor cell lines compared to the AXL-high-expressing tumor cell lines. These results, in the line of our previous study, indicated that AXL-low-expressing *EGFR*-mutated NSCLC cells were more sensitive to osimertinib than AXL-high-expressing *EGFR*-mutated NSCLC cells. However, regardless of AXL expression, a small population of tumor cells survived even after 72 h of exposure to 1 µmol/L osimertinib, suggesting osimertinib tolerance in these populations.

Next, we retrospectively assessed the correlations between AXL expression and the clinical efficacy of osimertinib administered as the first-line treatment in 29 patients with *EGFR*-mutated NSCLC. Expression of AXL in the cell cytoplasm of pre-EGFR-TKI-treated tumor samples was evaluated using immunohistochemistry (IHC) staining and scored as very high (3+), high (2+), low (1+), and no (0) expression of AXL (Supplementary Figure 1). Of the 29 *EGFR*-mutated NSCLC tumor specimens, high (2+ to 3+) and low (no to 1+) AXL expression was observed in 6 (21%), and 23 (79%) specimens, respectively. The response rate to osimertinib in patients with AXL-low expression (16/23; 70%) was higher than that observed in AXL-high expression (3/6; 50%), especially, in patients with AXL scores of 3+, the responder to osimertinib treatment was none (0/2; 0%) (Fig. 1c). In addition, osimertinib trended to cause tumor shrinkage more remarkably in patients with AXL-low expression compared to AXL-high expression ($p = 0.094$) (Fig. 1d). Notably, no complete response was achieved with osimertinib treatment in these 29 patients regardless of the AXL expression in tumors, indicating that small tumors remained as residual lesions with osimertinib treatment even in responders. These results suggested that in accordance with results in cell line experiments, though AXL-low-expressing *EGFR*-mutated NSCLCs are highly sensitive to osimertinib, drug tolerance following the acquisition of resistance always occurs.

To clarify the mechanisms of osimertinib tolerance, we performed receptor-tyrosine kinase (RTK) array analysis comparing AXL-low-expressing (HCC4006 and H3255) and AXL-high-expressing (PC-9) *EGFR*-mutated NSCLC cell lines treated with or without osimertinib for 72 h. Among 49 RTKs, IGF-1R phosphorylation was increased after osimertinib exposure in AXL-low-expressing tumor cell lines, but not in AXL-high-expressing tumor cell lines (Fig. 1e). Western blot analysis demonstrated that while the evaluated *EGFR*-mutated NSCLC cell lines expressed various levels of IGF-1R protein, AXL-low-expressing tumor cells reported higher levels of phosphorylated IGF-1R (pIGF-1R) compared to AXL-high tumor cell lines (Fig. 1b). Consistent with the results of RTK array, osimertinib exposure for 72 h enhanced pIGF-1R in AXL-low-expressing tumor cell lines (HCC4006, HCC827, and H3255) (Fig. 1f). Interestingly, while IGF-1R phosphorylation was initially inhibited for 3–24 h, IGF-1R protein expression, as well as IGF-1R phosphorylation, increased 72 h after osimertinib exposure compared to control. Flow cytometric analysis confirmed the increased IGF-1R protein expression in osimertinib treated HCC4006 cells, but not in PC-9 cells (Supplementary Fig. 2). The phosphorylation of MET in H3255 cells and FGFR3 in HCC4006 cells appeared to be increased by osimertinib in the RTK array. However, after western blotting, we detected no discernible increase in phosphorylated MET or FGFR3 in H3255 or HCC4006 cells, respectively, treated with osimertinib for 72 h (Supplementary Fig. 3a). We further assessed the effect of the MET inhibitor (crizotinib) and FGFR inhibitor (BGJ398) on

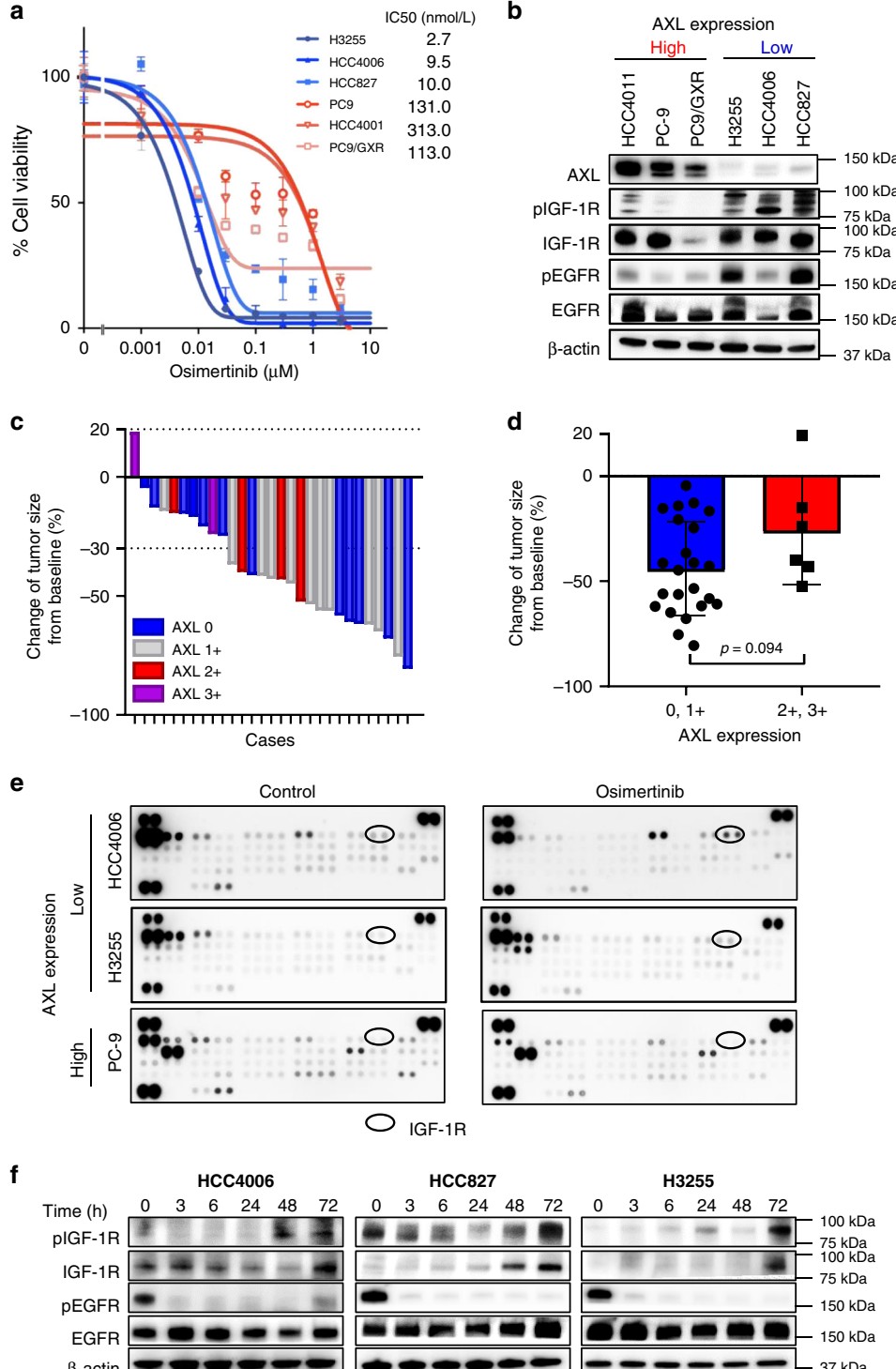

**Fig. 1 AXL-low tumor cells show high level of pIGF-1R after osimertinib exposure. a** *EGFR*-mutated NSCLC cells (seeded at $2 \times 10^3$ per well of a 96-well plate) were incubated with osimertinib at the indicated concentrations for 72 h. Cell viability was assessed using the Cell Counting Kit. Bars indicate s. d. of triplicate cultures. Data are presented as mean ± s.d. **b** Lysates of *EGFR*-mutated NSCLC cell lines were analyzed by western blotting. Data shown are representative of three independent experiments. **c** Correlation between the expression levels of the cytoplasmic AXL protein determined immunohistochemically, and response to treatment with osimertinib in *EGFR*-mutated NSCLC specimens from 29 patients. **d** Change in tumor size from baseline following osimertinib treatment in *EGFR*-mutated NSCLC patients with AXL-low ($n = 6$) and AXL-high ($n = 23$) expression. The data are expressed as mean and s.d. $p$ value is provided (two-sided Student's $t$-test). **e** Human tyrosine kinase phosphorylation array analysis in *EGFR*-mutated NSCLC cell lines in the presence or absence of osimertinib (30 nmol/L for HCC4006 and H3255 cells; 1 μmol/L for PC-9 cells) for 72 h. The circles indicate IGF-1R. **f** *EGFR*-mutated NSCLC cell lines were treated with osimertinib (30 nmol/L for HCC4006 and H3255 cells, 300 nmol/L for HCC827 cells) for indicated times, and lysates were analyzed by western blotting. Data shown are representative of three independent experiments.

osimertinib sensitivity in H3255 and HCC4006 cells, respectively, using the MTT assay. Although crizotinib and BGJ398 inhibited the phosphorylation of MET and FGFR3 in H3255 and HCC4006 cells, respectively, neither crizotinib nor BGJ398 remarkably affected the osimertinib sensitivity of H3255 and HCC4006 cells (Supplementary Fig. 3b–e). These results indicate that MET and FGFR3 are unlikely to play predominant roles in mediating osimertinib sensitivity in the *EGFR*-mutated lung cancer cells tested, at least, in our experimental conditions.

Next, we evaluated the correlation between pIGF-1R expression and the clinical efficacy of osimertinib administered as the first-line treatment in 16 patients with AXL-low (no to 1+) expressing *EGFR*-mutated NSCLC. In the pre-EGFR-TKI-treated tumor samples, pIGF-1R expression was evaluated using IHC staining and scored as very high (3+), high (2+), low (1+), and no (0) expression of pIGF-1R (Supplementary Fig. 4a). Of the 16 specimens, high (2+ to 3+) and low (no to 1+) pIGF-1R expression was observed in 5 (31%) and 11 (69%) specimens, respectively. In patients with pIGF-1R-high expression, the response rate to osimertinib was 100% (5/5); however, in patients with pIGF-1R-low expression, the response rate was 73% (8/11) (Supplementary Fig. 4b). In addition, osimertinib-induced tumor shrinkage more markedly in patients with pIGF-1R-high expression when compared with pIGF-1R-low expression ($p = 0.011$) (Supplementary Figure 4c), consistent with the results of the experiments performed using *EGFR*-mutated NSCLC cell lines (Fig. 1a, b). These results suggested that while the levels of phosphorylated IGF-1R correlated with better response to osimertinib, further increase in the phosphorylation of IGF-1R in association with osimertinib-mediated increase in the protein expression of IGF-1R may play a pivotal role in the tolerance observed in AXL-low-expressing *EGFR*-mutated NSCLCs.

**IGF-1R supported the survival of AXL-low tumor cells**. To assess the role of IGF-1R in osimertinib tolerance, we knocked down IGF-1R using appropriate siRNAs. IGF-1R knockdown did not affect the viability of *EGFR*-mutated NSCLC cells tested in the absence of osimertinib. In the presence of suboptimal concentrations (30 nmol/L for HCC4006 and H3255 cell lines, and 300 nmol/L for other cell lines) of osimertinib, knockdown of IGF-1R discernibly inhibited the viability of AXL-low-expressing HCC4006, HCC827, and H3255 cells (Fig. 2a and Supplementary Fig. 5a), but not AXL-high-expressing PC-9 or HCC4011 cells (Fig. 2b). Western blots demonstrated that in AXL-low-expressing cell lines, osimertinib exposure inhibited the phosphorylation of EGFR, its adaptor proteins GRB2-associated-binding protein 1 (Gab1) and Src homology and collagen homology (Shc), and downstream signaling molecules AKT and ERK (Fig. 2c and Supplementary Fig. 5b). IGF-1R knockdown further inhibited the phosphorylation of these molecules under the osimertinib exposure, consistent with the results of the cell viability assay. To further confirm the role of IGF-1R, we established IGF-1R knockout HCC827 cells by CRISPR-CAS9 targeting two different sites of IGF-1R (KO1 and KO2) (Supplementary Fig. 6, Fig. 2d). All tested IGF-1R KO clones were markedly more sensitive to osimertinib when compared with parental HCC827 cells (Fig. 2e). Following osimertinib exposure, the IGF-1R knockout further inhibited the phosphorylation of Gab1, Shc, AKT, and ERK, consistent with findings observed using IGF-1R knockdown cells with siRNA for IGF-1R (Fig. 2f).

These results clearly indicated IGF-1R is involved in tolerance and supported the survival of AXL-low-expressing *EGFR*-mutated NSCLC cells after osimertinib exposure.

**Osimertinib up-regulated IGF-1R mRNA expression via FOXA1**. We next explored the mechanism by which the IGF-1R protein was increased following osimertinib exposure. Quantitative RT-PCR revealed that osimertinib exposure up-regulated the expression of IGF-1R mRNA (Supplementary Fig. 7a). The induction of IGF-1R following osimertinib exposure was prevented by cycloheximide at both the mRNA and protein levels (Supplementary Fig. 7a, b, c). Hence, we hypothesized that IGF-1R upregulation by osimertinib might require de novo protein synthesis of transcription factors that activate IGF-1R mRNA expression. In order to identify these candidate transcription factors, we attempted to discover the transcription factors that could bind to the regulatory region around the IGF-1R promoter and that were up-regulated following osimertinib exposure. From the public ChIP-seq data integrated by ChIP-Atlas[13], we selected 79 candidate transcription factors that could bind to the DNase I hypersensitivity site 1 (DHS1) around IGF-1R transcription start site (TSS) (Supplementary Fig. 8a). Then, we performed quantitative RT-PCR analysis to determine the candidate transcription factors upregulated by osimertinib exposure. Four transcription factors including BCL6, CEBPA, FOXA1, and NFE2 were up-regulated by more than twofold after osimertinib treatment in HCC827 cells (Supplementary Data 1). To investigate the involvement of these four candidates in *IGF-1R* mRNA upregulation, we examined the effects of BCL6, CEBPA, FOXA1 and NFE2 knockdown by each shRNA in osimertinib treated HCC827 cells (Fig. 3a). The knockdown of FOXA1, but not NFE2, BCL6, or CEBPA, inhibited IGF-1R mRNA upregulation induced by osimertinib (Fig. 3a). We confirmed the effect of FOXA1 knockdown on the inhibition of IGF-1R mRNA induction using three different shRNAs (Fig. 3b). In addition, FOXA1 knockdown inhibited the upregulation of both total IGF-1R and phosphorylated IGF-1R protein induced by osimertinib, but failed to affect the status of total EGFR and phosphorylated EGFR protein (Fig. 3c). These results indicated that FOXA1 was indispensable for the IGF-1R upregulation induced by osimertinib exposure in HCC827 cells. We next examined the effects of FOXA1 overexpression in osimertinib treated cells. In HCC827 cells, overexpression of FOXA1 increased the levels of IGF-1R mRNA, total IGF-1R, and phosphorylated IGF-1R protein in the presence or absence of osimertinib, but had no effect on total EGFR and phosphorylated EGFR protein (Fig. 3d, e). These results indicated the specific role of FOXA1 as a transcriptional activator of IGF-1R. Next, we examined the effects of FOXA1 knockdown or overexpression on osimertinib tolerance in HCC827 cells. The number of osimertinib tolerant colonies was reduced by knockdown of FOXA1 using three different shRNAs and was increased by FOXA1 overexpression (Fig. 3f). These results suggested that FOXA1 contributed to enhance the osimertinib tolerance in HCC827 cells. In contrast to IGF-1R expression results shown in Supplementary Fig. 4a, FOXA1 induction following osimertinib exposure was not influenced by cycloheximide treatment, indicating that FOXA1 upregulation by osimertinib does not require de novo protein synthesis (Fig. 3g). We hypothesized that pre-existing signaling proteins or pathways might be responsible for the induction of FOXA1 mRNA by osimertinib. Accordingly, we observed that osimertinib-dependent FOXA1 induction was significantly inhibited in the IGF-1R knockout HCC827 cell clones (Fig. 3h). These results suggested that IGF-1R protein was involved in the signal transduction activating FOXA1 mRNA expression following osimertinib exposure. Since there is a consensus binding site of FOXA1 in the DHS1 around TSS of the IGF-1R gene (Fig. 3i and Supplementary Fig. 8b), we performed a ChIP assay to examine whether osimertinib treatment-induced changes in the epigenetic status of IGF-1R gene. Osimertinib treatment-induced transcriptionally active histone modifications

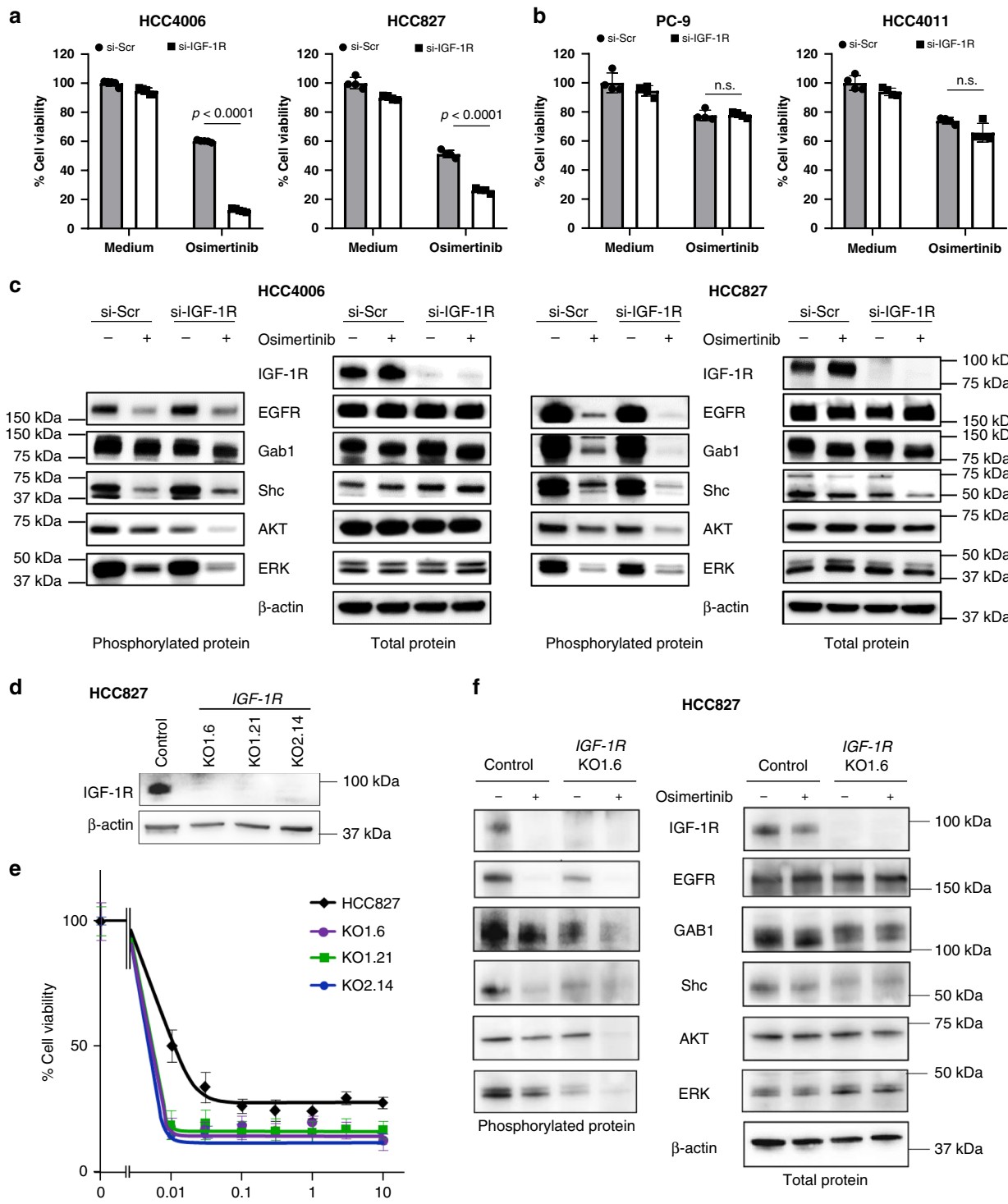

**Fig. 2 pIGF-1R supports the survival of AXL-low tumor cells after osimertinib exposure.** AXL-low expressing (**a**) and AXL-high expressing (**b**) *EGFR*-mutated NSCLC cell lines were treated with nonspecific control (si-Scr) or siRNA specific to IGF-1R (si-IGF-1R) for 72 h in the presence or absence of osimertinib (AXL-low of HCC4006 and HCC827:30 nmol/L and 300 nmol/L, respectively; AXL-high of PC-9 and HCC4011:1 µmol/L), and cell viability was determined. The percentage of growth is shown relative to untreated controls. Data are presented as mean ± s.d. Each sample was assayed in triplicate, with each experiment repeated at least three times independently. *p* values are provided (one-way ANOVA). n.s.: not significant. **c** si-Scr or si-IGF-1R was introduced into HCC4006 and HCC827 cells. After 24 h, the cells were incubated with or without osimertinib (30 nmol/L and 300 nmol/L, respectively) for 72 h and lysed, and the indicated proteins were detected by western blotting. **d** IGF-1R knockdown clones of HCC827 cells by CRISPR-CAS9 (KO-1-6, KO1-21, and KO2-14) were lysed and the proteins were detected by western blotting. **e** HCC827 and its IGF-1R knockdown clones were incubated with various concentrations of osimertinib, and cell viability was determined using the MTT assay. Data are presented as mean ± s.d. **f** HCC827 and KO1-6 clones were incubated with osimertinib (300 nmol/L) for 2 h, lysed, and the indicated proteins and their phosphorylation were detected by western blotting. Data shown are representative of three independent experiments.

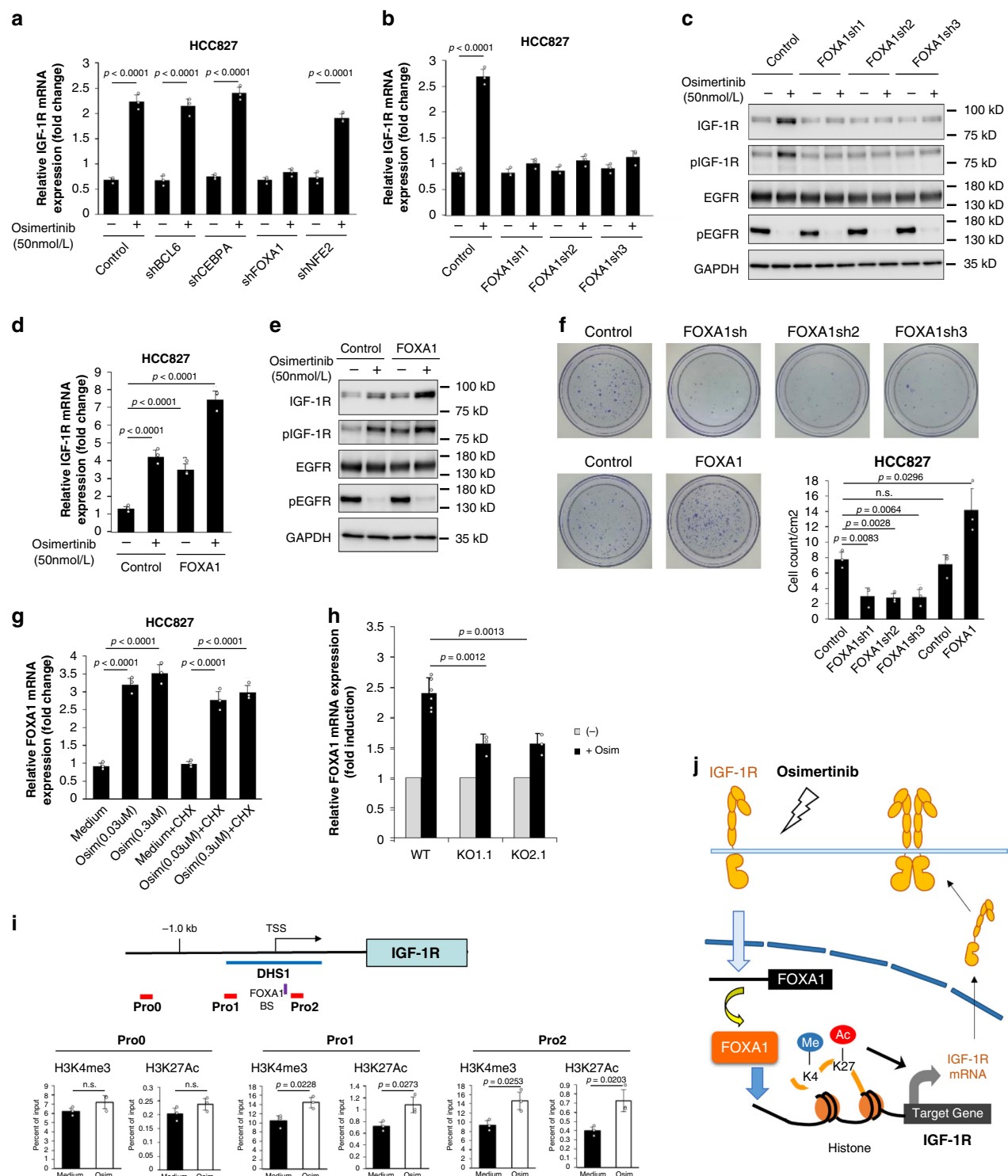

such as H3K4me3 and H3K27Ac within the DHS1 region (Pro1 and Pro2) but not outside (Pro0) (Fig. 3i). Collectively, these data suggested that osimertinib exposure activated FOXA1 expression through the signaling pathway comprising endogenous IGF-1R protein. Then, FOXA1 induced the transcriptionally more active epigenetic status of the IGF-1R gene, resulting in the positive feedback activation of IGF-1R in HCC827 cells (Fig. 3j).

**IGF-1R was associated with EGFR and adaptor proteins.** Next, we sought to clarify the mechanism by which IGF-1R

phosphorylation was augmented by osimertinib. IGF-1R is reportedly activated mainly by its ligand binding and interaction with other receptors, including EGFR[14]. We observed that the evaluated *EGFR*-mutated NSCLC cell lines produced very low levels of IGF-1 and IGF-2 proteins, the major ligands of IGF-1R, in the culture supernatant (Supplementary Fig. 9a). Moreover, the exogenous addition of a high concentration (50 ng/ml) of recombinant IGF-1, which could increase IGF-1R phosphorylation, did not remarkably affect osimertinib sensitivity in HCC4006, HCC827, or H3255 cells in vitro (Supplementary Fig. 9b, c). These results suggested that the IGF-1R ligands were

**Fig. 3 FOXA1 is involved in osimertinib-induced IGF-1R mRNA expression in HCC827 cells. a** Real-time quantitative polymerase chain reaction (qRT-PCR) analysis was performed to detect the expression of IGF-1R mRNA in HCC827 cells infected with lentiviruses expressing control shRNA (sh) or the shRNA for indicated molecules, with or without osimertinib treatment, for 24 h. **b** qRT-PCR of IGF-1R transcripts performed in HCC827 cells, similarly treated with osimertinib as in (**a**), introduced with three different shRNAs for FOXA1. **c** HCC827 cells with control or FOXA1 shRNAs were similarly treated with osimertinib, and the indicated proteins were detected by western blotting. **d** The expression of IGF-1R was detected by qRT-PCR in HCC827 cells infected with the control or the FOXA1 expressing retrovirus, following similar osimertinib treatment. **e** The indicated proteins were detected by western blotting in the indicated cells as in (**d**). **f** HCC827 cells with FOXA1 knockdown or overexpression were cultured for 18 days in the presence of osimertinib in a 60-mm dish. The dishes were stained with crystal violet, followed by imaging. The average number of drug-resistant colonies are presented in the right panel. **g** HCC827 cells were treated with osimertinib (Osim) and/or cycloheximide (CHX) (50 µg/mL) for 24 h. mRNA was harvested, and FOXA1 mRNA expression was evaluated by qRT-PCR. **h** qRT-PCR analysis was performed to detect the FOXA1 mRNA expression in HCC827 cells, as well as IGF-1R knockdown. **i** The promoter region of the IGF-1R gene is shown. The blue and purple bars represent the DNase I hypersensitivity site 1 (DHS1) and the consensus FOXA1 binding site (BS), respectively. The regions covered by primer sets used for ChIP assays are indicated as red bars, named Pro0 to Pro2. TSS: transcription start site. HCC827 cells with or without osimertinib (Osim) treatment for 24 h were cross-linked and the cell lysates were prepared. ChIP analyses of H3K4me3 and H3K27Ac on the indicated regions of the IGF-1R gene are shown below. **j** Schematic representation of the possible mechanism of osimertinib-induced expression of FOXA1 and IGF-1R. Me:methylation. Ac acetylation. Data shown are representative of three independent experiments. Data are presented as mean ± s.d. n.s. not significant. *p* values are provided (two-sided Student's *t*-test).

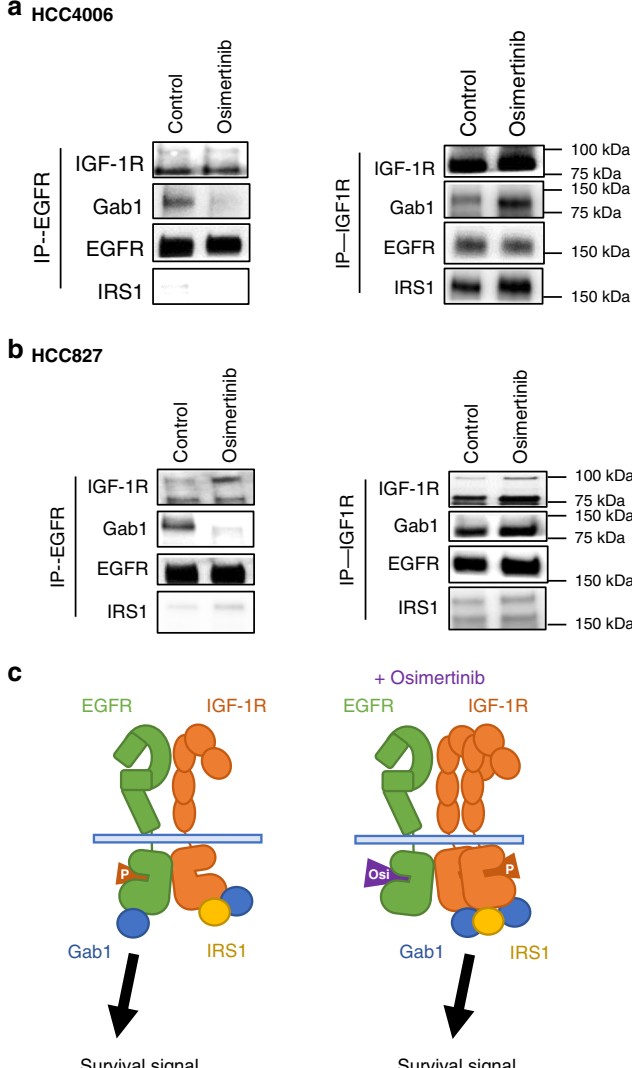

**Fig. 4 IGF-1R associated with EGFR and adaptor proteins, Gab1 and Shc.** HCC4006 (**a**) and HCC827 (**b**) cells treated with osimertinib (30 nmol/L and 300 nmol/L, respectively) for 72 h were lysed, and the indicated proteins were detected by western blotting, with immunoprecipitation of the indicated proteins. **c** Schema presenting the mechanism by which AXL-low- expressing *EGFR*-mutated NSCLC cells adapt to osimertinib exposure. Data shown are representative of three independent experiments. IP immunoprecipitation.

unlikely to be involved in IGF-1R phosphorylation induced by osimertinib exposure in these cell lines.

Therefore, we next examined the association of IGF-1R with EGFR and their adaptor proteins, such as Gab1, Shc, and insulin receptor substrate 1 (IRS1). Immunoprecipitation followed by western blotting demonstrated that EGFR constitutively associated with IGF-1R and Gab1, but not IRS1, in HCC4006 and HCC827 cells (Fig. 4a, b). Treatment with osimertinib did not affect the association of EGFR with IGF-1R; however, the association of EGFR with Gab1 decreased. Interestingly, IGF-1R also constitutively associated with its adaptor protein IRS1, and treatment with osimertinib trended to increase the association of IGF-1R with IRS1 and Gab1, but not EGFR.

Collectively, these findings suggested that EGFR and IGF-1R bind constitutively and to each protein associated with adaptor proteins, including Gab1, and may transduce the survival signal predominantly through EGFR in AXL-low-expressing *EGFR*-mutated NSCLC cells. Under osimertinib exposure, inactivated EGFR was still associated with IGF-1R, but no longer associated with Gab1. However, an increased amount of IGF-1R protein bound to EGFR, associated with adaptor proteins including IRS1 and Gab1, and could transduce survival signals in these cells (Fig. 4c).

**IGF-1R inhibitor inhibited the AXL-low tumor cell viability.** Next, we evaluated the effect of the small compound, linsitinib, which inhibits IGF-1R phosphorylation[15,16]. Our results demonstrated that 3 µmol/L or lower concentrations of linsitinib alone did not affect the viability of *EGFR*-mutated NSCLC cell lines, irrespective of the AXL expression (Supplementary Fig. 10). The combined use of 1 µmol/L linsitinib augmented the effect of osimertinib in AXL-low-expressing *EGFR*-mutated NSCLC cells (Fig. 5a and Supplementary Fig. 11a), but not in AXL-high-expressing *EGFR*-mutated PC-9 or HCC4011 cells (Fig. 5b). Similar linsitinib effects were observed in combination with gefitinib or dacomitinib, FDA approved first and second-generation EGFR-TKIs, respectively, in AXL-low-expressing *EGFR*-mutated NSCLC cell lines (Supplementary Fig. 12). Western blotting to explore the intracellular signaling demonstrated that osimertinib- or dacomitinib-alone inhibited phosphorylation of EGFR, Gab1, and AKT, but increased IGF-1R protein expression and IGF-1R phosphorylation in HCC4006, HCC827, and H3255 cells (Fig. 5c and Supplementary Fig. 11b). Longer treatment (7 days) with osimertinib and linsitinib eradicated HCC4006, HCC827, and H3255 cells (Supplementary Fig. 13). These results indicated that the IGF-1R inhibitor linsitinib could

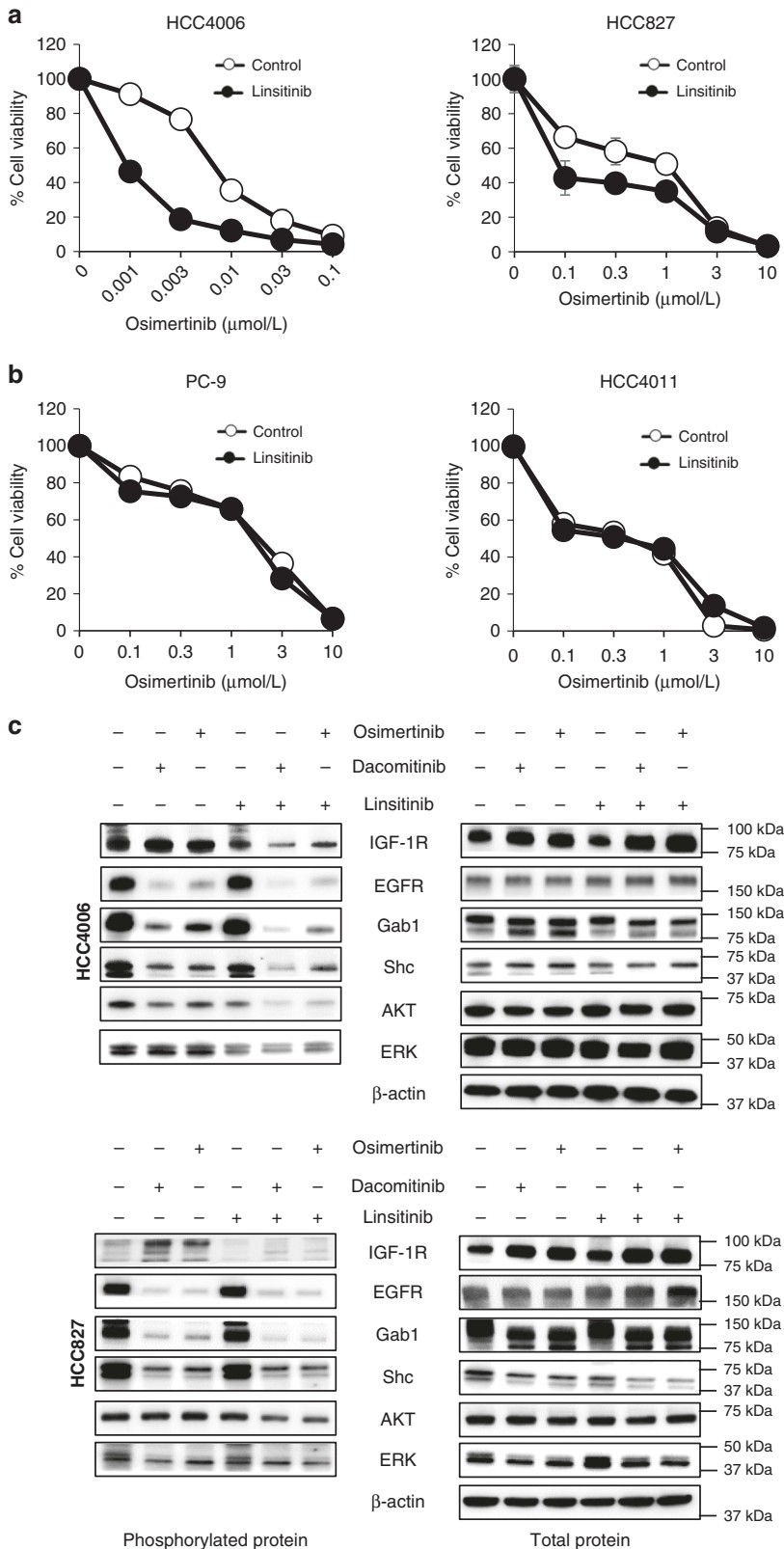

**Fig. 5 IGF-1R inhibitor inhibited the viability of AXL-low tumor cells exposed to osimertinib.** AXL-low expressing (**a**) and AXL-high expressing (**b**) *EGFR*-mutated NSCLC cell lines were treated with various concentrations of osimertinib for 72 h in the presence or absence of linsitinib (1 μmol/L), and cell viability was determined. The percentage of growth is shown relative to untreated controls. Data are presented as mean ± s.d. Each sample was assayed in triplicate, with each experiment repeated at least three times independently. **c** HCC4006 and HCC827 cells were treated with osimertinib (30 and 300 nmol/L, respectively), dacomitinib (30 and 300 nmol/L, respectively), and/or linsitinib (1 μmol/L). After 72 h, the cells were lysed, and the indicated proteins were detected by western blotting. Data shown are representative of three independent experiments. Bars indicate s.d.

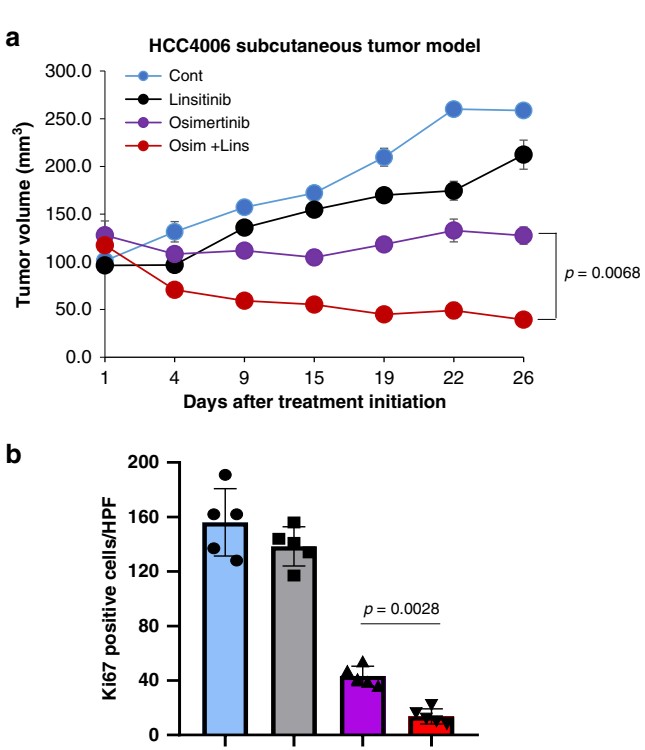

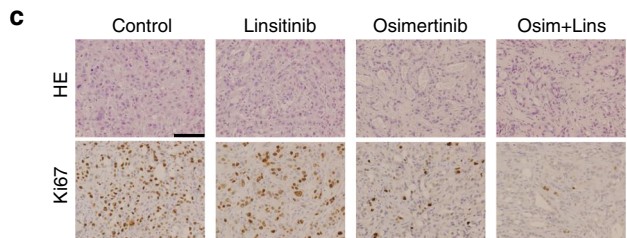

**Fig. 6 Linsitinib with a suboptimal dose of osimertinib regresses AXL-low tumors in vivo. a** HCC4006 cell line-derived xenograft (CDX) tumors were treated with vehicle (control: n = 4 mice), linsitinib 50 mg/kg (n = 4 mice), osimertinib 5 mg/kg (n = 5 mice), or linsitinib 50 mg/kg plus osimertinib 5 mg/kg (n = 5 mice), administered daily by oral gavage. Tumor volumes were measured over time from the start of treatment (mean ± s.e.m.). p values are provided (two-sided Student's t-tests). **b** Quantification of proliferating cells, as determined by their Ki-67-positive proliferation index (percentage of Ki-67-positive cells) as described in "Methods". Columns, mean of five areas. Data are presented as mean ± s.d. p values are provided (two-sided Student's t-tests). HPF high power field. **c** Representative images of HCC4006 xenografts for H&E staining and immunohistochemical staining with antibodies to human Ki-67. Bar, 100 μm.

prevent osimertinib tolerance in AXL-low-expressing *EGFR*-mutated NSCLC cells.

**Transient linsitinib combo cured AXL-low tumors in vivo**. We assessed the effect of linsitinib combined with osimertinib in vivo. Subcutaneous tumors in HCC4006 cells grew rapidly and linsitinib monotherapy did not inhibit the growth. Osimertinib treatment at a suboptimal dose (5 mg/kg)[17] prevented the enlargement of the tumors. Combined treatment with osimertinib and linsitinib induced tumor shrinkage (Fig. 6a). These treatments did not affect the body weight in mice (Supplementary Fig. 14a), indicating the treatment feasibility. IHC analyses

showed that the osimertinib treatment decreased the number of proliferating tumor cells, and the combined use of linsitinib with osimertinib further suppressed it (Fig. 6b, c).

While recent clinical trials reported the feasibility of linsitinib combined with the EGFR-TKI erlotinib, the continuous administration of linsitinib resulted in higher rates of erlotinib dose interruptions, reduction, and discontinuation[18]. Hence, we next evaluated the effect of transient combination with linsitinib in subcutaneous tumors in HCC827 or HCC4006 cells. Subcutaneous tumors in HCC827 or HCC4006 cells grew rapidly and linsitinib monotherapy did not inhibit the growth. Osimertinib treatment at a higher dose (12.5 mg/kg) decreased the tumor size, but small tumors remained during osimertinib treatment. After the discontinuation of osimertinib treatment, these tumors re-grew (Fig. 7a, b). The transient combination of linsitinib for 10 days with continuous osimertinib treatment resulted in the disappearance of tumors. Surprisingly, these tumors never re-grew even after the discontinuation of osimertinib treatment. No remarkable loss of body weight was observed in each treatment group (Supplementary Fig. 14b).

In addition, we evaluated the transient combination with linsitinib in a PDX model (Fig. 7c). We established two PDXs from patients with *EGFR* mutations[19]. PDX LC#7 had *EGFR*-L858R and a discernible level of AXL, lacking phosphorylated IGF-1R. On the other hand, PDX LC#11 had *EGFR*-exon 19 deletion and discernible levels of phosphorylated IGF-1R, but not AXL (Fig. 7d). Therefore, we chose PDX LC#11 for this set of experiments. The PDX LC#11 tumors grew gradually and linsitinib monotherapy did not inhibit the growth. Osimertinib treatment at a higher dose (12.5 mg/kg) decreased the tumor size, but small tumors persisted with osimertinib treatment. After the discontinuation of osimertinib treatment, these tumors re-grew rapidly (Fig. 7e). The transient combination of linsitinib with continuous osimertinib treatment for 10 days resulted in the disappearance of tumors. These tumors hardly re-grew even after the discontinuation of osimertinib treatment. These results suggested that the transient combination of linsitinib with continuous osimertinib treatment could cure or dramatically delay tumor recurrence in AXL-low-expressing *EGFR*-mutated NSCLC.

## Discussion

The heterogeneity of cancer cells limits the efficacy of cancer treatment[20,21]. Recent studies uncovered that lung cancers with classical *EGFR* mutations, such as exon 19 and exon 21 L858R mutation, are a heterogeneous disease in terms of response to EGFR-TKI treatment[22,23]. The heterogeneous responses observed could be caused by various factors, including the presence of compound mutations in *EGFR*[24], expression of resistance proteins such as AXL and MET in tumor cells[25,26], and variety of stroma-derived resistance molecules such as hepatocyte growth factor (HGF)[27]. The present study revealed that AXL-high expression was associated with a poor initial response to the third-generation EGFR-TKI osimertinib. On the other hand, while AXL-low-expressing tumor cells are highly sensitive to osimertinib, a small population of these cells demonstrates tolerance due to the increased expression and phosphorylation of IGF-1R. Moreover, transient inhibition of IGF-1R with osimertinib could lead to the eradication of *EGFR*-mutated lung cancer cells.

We previously reported that the susceptibility of *EGFR*-mutated lung cancer cells to EGFR-TKIs including osimertinib inversely correlated with the expression of AXL in tumor cells[12]. In the present study, we confirmed this and further observed that while a small population of both AXL-high and AXL-low-

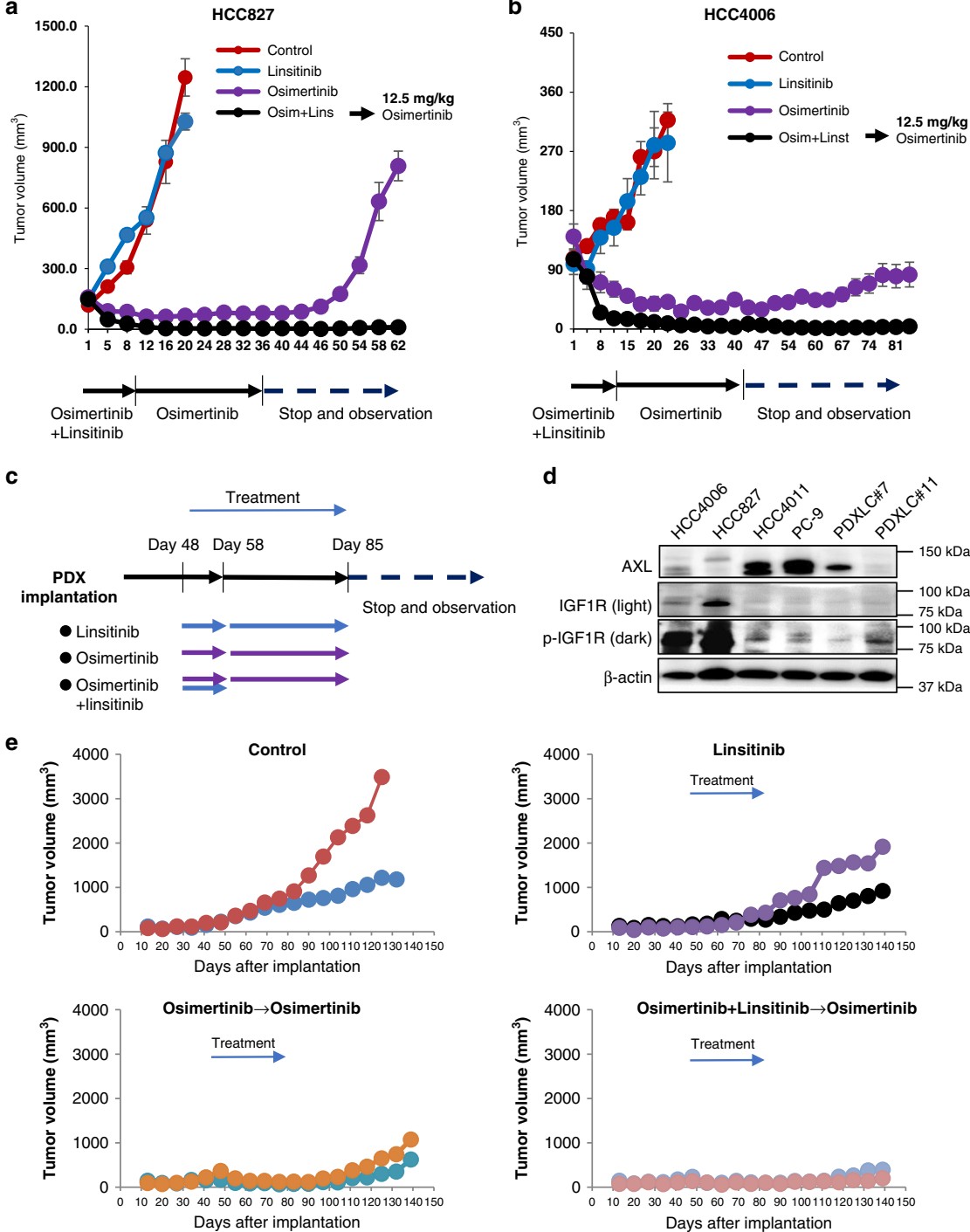

**Fig. 7 Transient combination of linsitinib with osimertinib cured AXL-low tumors in vivo.** The mice with HCC827 (**a**)- or HCC4006 (**b**)-cell line-derived xenograft (CDX) tumors were treated with vehicle (control:*n* = 4 mice), linsitinib 50 mg/kg (*n* = 4 mice), osimertinib 12.5 mg/kg (*n* = 5~6 mice), or linsitinib 50 mg/kg plus osimertinib 12.5 mg/kg (*n* = 5 mice), administered by oral gavage for 10 days. Next, the groups administered osimertinib and linsitinib plus osimertinib were treated with osimertinib 12.5 mg/kg daily for an additional 25 days. Thereafter, the treatment was terminated and tumor regrowth was evaluated by day 62. Tumor volumes were measured over time from the start of treatment (mean ± s.e.m). **c** Treatment schedule for the PDX model. Forty-eight days after implantation of PDX tumors, the mice with PDX tumors were treated with vehicle (control), linsitinib 50 mg/kg, osimertinib 12.5 mg/kg, or linsitinib 50 mg/kg plus osimertinib 12.5 mg/kg, administered by oral gavage for 10 days. Next, groups administered osimertinib and linsitinib plus osimertinib were treated with osimertinib 12.5 mg/kg daily for an additional 27 days. The linsitinib group was treated with linsitinib 50 mg/kg daily for an additional 27 days. Thereafter, the treatment was terminated and tumor regrowth was evaluated by day 140. **d** Lysates of cell lines and PDXs were analyzed by western blotting. Data shown are representative of two independent experiments. **e** PDX LC#11 (*n* = 2 mice/group) were treated as shown in (**c**).

expressing tumor cells showed tolerance to osimertinib exposure and become the base of acquired resistance, the mechanisms of the tolerance largely differed between AXL-high and AXL-low-expressing tumor cells. In AXL-high-expressing *EGFR*-mutated lung cancer cells, osimertinib exposure inhibits ERK phosphorylation and thereby decreases the expression of SPRY4 maintained by ERK-mediated MAPK signal[12,28,29]. Since SPRY4 inhibits activation of various receptor-tyrosine kinases including AXL, decreased expression of SPRY4 by osimertinib exposure results in the activation of AXL protein without affecting the level of AXL protein expression and restoring the survival signal mainly via AKT[12]. On the other hand, we demonstrated that in AXL-low-expressing *EGFR*-mutated lung cancer cells, osimertinib exposure increased protein expression and phosphorylation of IGF-1R and thereby restored the survival signal mainly via Gab1 and IRS1 to emerge tolerant. Interestingly, the AXL-low-expressing tumor cells, which were more sensitive to osimertinib had a higher baseline level of phosphorylated IGF-1R than the AXL-high-expressing tumor cells, suggesting that phosphorylated IGF-1R at baseline level might have less impact on osimertinib sensitivity than AXL expression. Nevertheless, in the AXL-low-expressing tumor cells, through increased protein expression and phosphorylation of IGF-1R by osimertinib exposure (e.g., for 3 days), tolerance emerged via epigenetic modification. This increased expression of the transcription factor FOXA1 and restored the survival signal via IGF-1R associated with Gab1 and IRS1. The reasons why Gab1/IRS1 signaling is caused by increased IGF-1R phosphorylation via FOXA1 by osimertinib exposure but not baseline IGF-1R phosphorylation remains to be elucidated. Furthermore, the reason why baseline level expression of phosphorylated IGF-1R is higher in the AXL-low-expressing tumor cells than that in the AXL-high-expressing tumor cells is also unclear at present. Further examinations are warranted to clarify these mechanisms in the future.

Reportedly, it has been demonstrated that IGF-1R plays a crucial role in the emergence of cells tolerant to EGFR-TKIs. First-generation EGFR-TKIs stimulate the expression of KDM5A, a histone demethylase, resulting in increased IGF-1R phosphorylation associated with IGF-BP3 overexpression, and thereby inducing drug-tolerant cells[11]. The third-generation EGFR-TKIs, PF299804, and WZ4002, activated IGF-1R owing to a loss of IGFBP3 expression via methylation of its promotor lesions[30]. In the present study, we demonstrated the novel findings that tolerance to the clinically available third-generation EGFR-TKI osimertinib is induced by increased IGF-1R protein expression. Osimertinib exposure activated expression of transcription factor FOXA1 through the signaling pathway comprising endogenous IGF-1R protein, with FOXA1 then inducing the transcriptionally more active epigenetic status of the IGF-1R gene, resulting in the positive feedback activation of IGF-1R (Fig. 3j). FOXA1 is a pioneering transcription factor that can bind to and open "closed chromatin" and trigger transcriptional events on target genes[31–33]. A previous study reported that FOXA1 was frequently mutated or amplified in various types of cancer[34–36]. In some cases of lung cancer, FOXA1 is overexpressed as a result of gene amplification[37]. These studies suggested that FOXA1 contributed to oncogenesis in human lung cancer. However, it remains unclear whether FOXA1 is involved in drug resistance in lung cancer. Here, we demonstrated the indispensable roles of FOXA1 on the emergence of cells tolerant to osimertinib via IGF-1R activation associated with increased mRNA and protein expression of IGF-1R in AXL-low-expressing *EGFR*-mutated NSCLC cells following osimertinib exposure.

Crosstalk between IGF-1R and other RTKs, including EGFR, HER2, VEGFR2, PDGFR, MET, and ALK, results in reciprocal compensatory mechanisms that limit response and/or mediate acquired resistance to treatments that target an individual pathway[14]. It has been recently reported that EGFR and IGF-1R can interact on multiple levels, either through a direct association between the two receptors, by the availability of respective ligands, or indirectly via the common interaction partners such as G protein-coupled receptors or downstream signaling molecules including adaptor proteins[14]. Our data suggest that IGF-1R is likely to be activated by crosstalk via direct association with EGFR rather than its ligand binding (Fig. 4). In AXL-low-expressing tumor cells, EGFR and IGF-1R bind constitutively and to each protein associated with their adaptor proteins, including Gab1 (Fig. 4). While osimertinib decreased the association of EGFR with Gab1, it did not inhibit the association of IGF-1R with EGFR or Gab1. Together with findings that osimertinib increased the IGF-1R protein level, these results strongly suggest that an increased amount of IGF-1R, which maintains the association with EGFR and Gab1, may restore survival signals in AXL-low-expressing *EGFR*-mutated lung cancer cells exposed to osimertinib. This crosstalk and its involvement in osimertinib tolerance provide a clear rationale for the dual targeting of EGFR and IGF-1R.

On the other hand, IGF-1R plays an indispensable role in homeostasis, and therefore its continuous inhibition may cause various adverse reactions including the poor control of blood glucose levels[38,39]. Rociletinib, a third-generation EGFR-TKI, has known metabolite activity to inhibit IGF-1R[40]. Although rociletinib showed similar anti-tumor efficacy with osimertinib in T790M-positive *EGFR*-mutated lung cancer, its safety profile was inferior compared to osimertinib[41]. Considering these issues, we chose the transient combination of the IGF-1R inhibitor with osimertinib, demonstrating a favorable efficacy and safety in the CDXs and PDX models of AXL-low-expressing *EGFR*-mutated lung cancer. These novel findings may be crucial in considering therapeutic strategies that cure or dramatically improve the prognosis of *EGFR*-mutated lung cancer. Since osimertinib is a selective inhibitor of mutated *EGFR* in tumor cells and spares wild type EGFR expressed mainly in host cells, the feasibility of a transient combination of IGF-1R inhibitor with osimertinib may be more superior to first or second generation EGFR-TKIs.

In conclusion, we uncovered the mechanism by which AXL-low-expressing *EGFR*-mutated lung cancer cells demonstrated tolerance to osimertinib. Our results indicate that optimal inhibition of the tolerant signal via IGF-1R combined with osimertinib may dramatically improve the outcome of AXL-low-expressing *EGFR*-mutated lung cancer. Furthermore, the safety and efficacy of the transient combination of IGF-1R inhibitor and osimertinib should be evaluated in the clinical trials.

## Methods

**Cell cultures and reagents**. Six human NSCLC cell lines with mutations in *EGFR* were utilized. The human NSCLC cell lines HCC4011 and H3255 were generously provided by Dr. David P. Carbone (Ohio State University Comprehensive Cancer Center, Columbus, OH) and Dr. John D. Minna (University of Texas Southwestern Medical Center, Dallas, TX), respectively. The human cell lines HCC827 and HCC4006 were purchased from the American Type Culture Collection (Manassas, VA), and the PC-9 cell line was obtained from the RIKEN Cell Bank (Ibaraki, Japan). The PC-9GXR cells, which contain deletions in the *EGFR*-exon 19 and the T790M mutation, were established at Kanazawa University (Kanazawa, Japan) from the PC-9 cell xenograft tumors in nude mice that had acquired resistance to gefitinib[12]. All of these cell lines were maintained in Roswell Park Memorial Institute (RPMI) 1640 medium (GIBCO, Carlsbad, CA) with 10% fetal bovine serum (FBS), penicillin (100 U/mL), and streptomycin (50 g/mL) in a humidified $CO_2$ incubator at 37 °C. All cells were passaged for <3 months before being renewed with frozen, early passage stocks. Cells were regularly screened for mycoplasma using a MycoAlert Mycoplasma Detection Kit (Lonza). Cell lines were authenticated by DNA fingerprinting. Osimertinib, dacomitinib, gefitinib, and linsitinib were obtained from Selleckchem (Houston, TX).

**Cell viability assay**. Tumor cells ($2-3 \times 10^3$ cells/100 μL/well) in RPMI 1640 medium supplemented with 10% FBS were plated in 96-well plates and cultured with the indicated compound for 72 h. After culturing, cell viability was measured using a CCK-8 kit (Dojindo Laboratories). The percentage of growth was determined relative to the untreated controls. Experiments were repeated at least three times with triplicate samples.

**Human phospho-kinase antibody array**. The relative phosphorylation levels of 49 kinases and two related total proteins were measured using the Human Phospho-Kinase Array Kit (R&D Systems), using a modification of the manufacturer's instructions. In brief, cells were cultured in RPMI-1640 containing 10% FBS and lysed in the array buffer prior to reaching confluence. The arrays were blocked with a blocking buffer and incubated with 450 μg of the cell lysate overnight at 4 °C. The arrays were washed, incubated with a horseradish peroxidase (HRP)-conjugated phospho-kinase antibody, and treated with SuperSignal West Dura Extended Duration Substrate Enhanced Chemiluminescent Substrate (Pierce Biotechnology, Rockford, IL). Each experiment was independently performed at least twice.

**Antibodies and western blotting**. Protein aliquots of 25 μg each were resolved by SDS polyacrylamide gel electrophoresis (Bio-Rad, Hercules, CA) or 1000 μg aliquots of total proteins were immunoprecipitated with the appropriate antibodies. The immune complexes were recovered with Protein G-Sepharose or Protein A-Sepharose beads (Zymed Laboratories, California). Electrophoresed protein samples or immunoprecipitated samples were transferred to polyvinylidene difluoride membranes (Bio-Rad). After washing three times, the membranes were incubated with blotting-grade blocker (Bio-Rad) for 1 h at room temperature and overnight at 4 °C with primary antibodies to p-AXL (Tyr702), t-AXL, p-EGFR, p-IGF-1R, t-IGF-1R, p-Akt (Ser473), t-Akt, p-Gab1, t-Gab-1, p-Shc, t-Shc, IRS-1,β-actin (13E5) (1:1,000 dilution; Cell Signaling Technology, Danvers, MA, USA), p-Erk1/2 (Thr202/Tyr204), t-Erk1/2, and t-EGFR (1:1000 dilution, R&D systems).

After washing three times, the membranes were incubated 1 h at room temperature with HRP-conjugated species-specific secondary antibody. Immunoreactive bands were visualized using SuperSignal West Dura Extended Duration Substrate Enhanced Chemiluminescent Substrate (Pierce Biotechnology). Each experiment was independently performed at least three times.

**Patients**. Tumor specimens containing *EGFR*-activating mutations, prior to the initial treatments with osimertinib as the first line treatment, were obtained from 29 non-small cell lung cancer patients hospitalized at the University Hospital, Kyoto Prefectural University of Medicine (Kyoto, Japan), Nagasaki University Hospital (Nagasaki, Japan), International Medical Center, Saitama Medical University (Saitama, Japan), or the National Hospital Organization Kinki-chuo Chest Medical Center (Osaka, Japan). The study protocol was approved by the Institutional Review Boards of the Kyoto Prefectural University of Medicine, Nagasaki University, Saitama Medical University, and National Hospital Organization Kinki-chuo Chest Medical Center. All patients provided written informed consent.

**Histological analyses of tumors**. In brief, the formalin-fixed, paraffin-embedded tissue sections (4-μm thick) were deparaffinized. The antigen was retrieved by microwaving the tissue sections in 10 mM citrate buffer (pH 6.0). Proliferating cells were detected by incubating the tissue sections with the Ki-67 antibody (Clone MIB-1; DAKO Corp, Glostrup, Denmark). Based on the expression patterns, tumor cells in tissue specimens were separately evaluated for the expression of AXL using an anti-AXL antibody (1:200; goat polyclonal, R&D SYSTEMS) and pIGF-1R using an anti- Phospho-IGF1- R antibody (1:80; Cell Signaling Technology, Danvers, MA, USA). Since immunohistochemical studies have shown that AXL and pIGF-1R are present primarily in the cytoplasm of cells and that its staining varies in intensity, we quantified its expression as negative (0), weak (1+), moderate (2+), and strong (3+) compared to vascular endothelial cells as an internal control[12]. After incubation of the specimens with the secondary antibody and treatment using the Vectastain ABC Kit (Vector Laboratories, Burlingame, CA), the peroxidase activity was visualized using 3,3′-diaminobenzidine (DAB) as a chromogen. Next, the sections were counterstained with hematoxylin.

**Quantification of immunohistochemistry results**. The five areas containing the highest numbers of positively stained cells within each section were selected for histologic quantitation using light or fluorescent microscopy at a 400 fold magnification.

**Flow cytometry**. For EGFR detection, single-cell suspensions of tumor cells ($5 \times 10^5$) were treated on ice with or without Brilliant Violet 421TM anti-human EGFR antibody (Biolegend Cat.No.352911, 2.5 μl/sample) for 30 min. For IGF-1R detection, single-cell suspensions of tumor cells ($5 \times 10^5$) were pre-treated with PerFix-nc Kit (Beckman Coulter, Cat.No.B31167, 50 μL 2% FBS, and 5 μL Buffer 1 per sample) at room temperature for 15 min for membrane permeabilization. After washing, resultant cells were incubated with 300 μL of Buffer 2 and 1 μL of IGF-1 Receptor B (D23H3) rabbit monoclonal antibody per sample, at room temperature for 30 min. Then, the cells were washed and treated with 300 μL of Buffer 2 and

1 μL of Alexa-488-conjugated antibody per sample, at room temperature for 30 min. After washing twice, the cells were resuspended in 500 μL of the Final buffer per sample and analyzed using a BD FACSCANTOII system.

**Transfection of siRNAs**. Duplexed Silencer® Select siRNA for *IGF-1R* was purchased from Invitrogen (Carlsbad, CA) and transfected into cells using Lipofectamine RNAiMAX (Invitrogen) in accordance with the manufacturer's instructions. In all experiments, Silencer® Select siRNA for Negative Control no.1 (Invitrogen) was used as the scrambled control. The sense strand sequences of the oligonucleotides used for pLKO.1 construct specific to BCL6, CEBPA, FOXA1, and NFE2 are listed in Supplementary Table 1a.

**Generation of IGF-1R knockout HCC827 cells by CRISPR/Cas9 system**. *IGF-1R*-specific single guide RNAs (sgRNAs) were designed using CRISPRdirect (http://crispr.dbcls.jp). To generate *IGF-1R* knockout cells, we used two pairs of sgRNAs. The sequences of sgRNA #1 and #2 for the partial deletion of *IGF1R* exon 2 (KO1) are 5′-GCGTTGCGGATGTCGATGCC-3′ and 5′-GCGGTAGCTGCGGTAGTCC T-3', respectively. The sgRNAs #3 and #4 for the partial deletion of *IGF1R* exon 9 (KO2) are 5′-GCTTCTCAGTTAATCGTGAAG-3′ and 5′-GAGCAGTAATTGT GCCGGTAA-3′. Each sgRNA was cloned into pX330-U6-Chimeric_BB-CBh-hSpCas9, a gift from Dr. Feng Zhang (Addgene plasmid #42230; http://n2t.net/addgene:42230; RRID: Addgene_42230), followed by their general cloning protocol[42].

HCC827 cells in 6-well plate were transfected with the plasmids using 1 mg/ml PEI MAX (Polysciences, Inc.). For each well, 150 ng of the two CRISPR/Cas9 plasmids and 50 ng of pPurΔDTA plasmid encoding puromycin resistant gene (gifted by Dr. Ryu Imamura, Kanazawa Univ., Japan) were used. After puromycin selection, drug-resistant cells were diluted and passaged, and several single colonies were picked up for further experiments. To examine the genotypes of the cell clones, PCR was performed using Tks Gflex DNA Polymerase (Takara Bio Inc.) with the following primers: 5′-ATGGTCGGTTGGAGTGTGTTG-3′ and 5′-CAC TCGGAACAGCAGCAAGTAC-3′ for KO1 cells, 5′-TGCCAGAGTATCTGATA GCCTGAC-3′ and 5′-TAGGGCTCAGGCACATTACAAC-3′ for KO2 cells.

**Prediction of transcription factors for IGF-1R expression**. In order to obtain candidate transcription factors that regulate IGF-1R expression, we first searched the DNase I hypersensitivity sites (DHS) regions around the IGF-1R transcription start site (TSS) based on the prediction by the public DNase-seq peaks using ChIP-Atlas[13]. In brief, we selected H. sapiens (organism), DNase-seq (Antigen Class), All cell types (Cell type Class) and 100 (Threshold for Significance) in ChIP-Atlas Peak Browser (https://chip-atlas.org/peak_browser) to visualize the DNase-seq data with the Integrative Genomics Viewer (IGV). We identified two potential DHS regions (DHS1: chr15:99191191-99192070 and DHS2: chr15:99190087-99191190, hg19) around the TSS of the IGF-1R gene (Supplementary Fig. 8a). We next searched the CTCF binding motif using CTCFBSDB 2.0 (http://insulatordb.uthsc.edu/)[43]. Since the DHS2 region contains several potential CTCF binding sites and may function as an insulator region, we focused on the candidate transcription factors which bind to the DHS1 region. From the public ChIP-seq peaks obtained in ChIP-Atlas Peak Browser as described above except "DNase-seq" was replaced by "TFs and others", we observed 79 candidate transcription factors (listed in Supplementary Data 1) that have binding peaks in the DHS1 region.

**Quantitative RT-PCR**. For RNA quantification, reverse transcription of the collected RNAs was performed using SuperScript VILO cDNA synthesis Kit and Master Mix (Invitrogen). Quantitative PCR analysis was performed with FastStart Universal SYBR Green Master (Roche) using ViiA7 Real-Time PCR system (Applied Biosystems)[44]. PCR data were normalized with respect to the control human GAPDH expression. The averages from at least three independent experiments are shown with the standard deviations. *P*-values were calculated between control and the samples using the Student's *t*-test. Primers used for the quantitative PCR are listed in Supplementary Table 1b.

**Chromatin immunoprecipitation (ChIP) assays**. ChIP experiments were performed as follows[44]. In brief, cells were cross-linked with 1% paraformaldehyde for 10 min and then incubated with 200 mM glycine for 5 min to quench reactive aldehydes. The cross-linked chromatins were fragmented by Bioruptor II ultra-sonicator (BM Equipment Co., Japan), and immunoprecipitated with rabbit antibody (anti-H3K4me3 (#07-473, Millipore) and anti-H3K27Ac (#39133, Active motif)) bound to Dynabeads Protein G (Invitrogen). The immunoprecipitated DNA was extracted sequentially with phenol and chloroform, and recovered with ethanol precipitation. The enrichment of the specific amplified region was analyzed by quantitative PCR and the percentage enrichment of each modification over input chromatin DNA was shown. Primers used for the quantitative PCR are listed in Supplementary Table 1c.

**CDX models**. Suspensions of $5 \times 10^6$ cells were injected subcutaneously into the flanks of six-week-old SHO mice (Crlj:SHO-Prkdc^scid^Hr^hr^, Charles River, Yokohama, Japan). Once the mean tumor volume reached ~100-300 mm³, the mice

were orally administered the targeted drugs and their body weight and general conditions were monitored daily. Tumors were measured twice weekly using calipers and their volumes were calculated as $width^2 \times length/2$. The study protocol was approved by the Ethics Committee on the Use of Laboratory Animals and the Advanced Science Research Center, Kanazawa University, Kanazawa, Japan (approval no. AP-122505). In accordance with the institutional guidelines, surgery was performed after the animals were anesthetized with sodium pentobarbital and all efforts were made to minimize animal suffering.

**PDX models**. Patient tumor samples were obtained with informed consent. Surgically resected tumor specimens were divided into small pieces (3–5 mm) and implanted into the subcutaneous flank tissue of SHO mice[19]. The PDX LC#7 was from an 81-year-old Japanese male (pT2N2M0, stage IIIA) with lung adenocarcinoma containing *EGFR*-L858R. The PDX LC#11 was from a 69-year-old Japanese male (pT2aN1M1a, stage IV) with lung adenocarcinoma containing the EGFR exon19 deletion[19]. The study protocol was approved by the Ethics Committee on the Use of Laboratory Animals and the Advanced Science Research Center, Kanazawa University, Kanazawa, Japan.

**Statistical analysis**. Data from the MTT assays and tumor progression of xenografts are expressed as means ± standard deviation (SD) and as means ± standard error (SE), respectively. The statistical significance of differences was analyzed using one-way ANOVA and Spearman rank correlations. Progression-free survival (PFS) and 95% confidence intervals (CIs) were determined using the Kaplan–Meier method and compared using the log-rank test. Hazard ratios (HRs) of clinical variables for PFS were determined using a univariate Cox proportional hazards model. All statistical analyses were performed using GraphPad Prism Ver. 6.0 (GraphPad Software, Inc., San Diego, CA, USA), with a two-sided *P*-value < 0.05 considered statistically significant.

**Reporting summary**. Further information on research design is available in the Nature Research Reporting Summary linked to this article.

## Data availability

All relevant data are included in the paper and its supplementary information files. The source data for Fig. 3i and Supplementary Fig. 8a, b are available at https://chip-atlas.org/peak_browser. Source data are provided with this paper.

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

## Acknowledgements

We thank Dr Ryu Imamura (Kanazawa University, Japan) for technical assistance and providing CRISPR/Cas9 plasmids. This work was supported by JSPS KAKENHI Grant Number JP16H05308 (to S.Y.), the Project for Cancer Research And Therapeutic Evolution (P-CREATE) from the Japan Agency for Medical Research and development, AMED, Grant Number 16cm0106513h0001 (to S.Y.), research grants from Boehringer-Ingelheim, Eisai, and MSD, and Extramural Collaborative Research Grant of Cancer Research Institute, Kanazawa University.

## Author contributions

Conception and design: T. Yamada, S.Y. Development of methodology: M.T., T.S., T. Yamano. Acquisition of data: R.W., K. Kita, S.A., K.F., M.T., T. Yamano, H.U., R.H., T.S., X.H., and K.M. Analysis and interpretation of data: W.R., T. Yamada, W.W., and S.Y. Writing of the manuscript: W.R., T.S., and S.Y. Administrative, technical, or material support: T. Yamada, H.T., A.N., A.T., S.T., K.O., K. Yamashita, A.Y., K.T., K. Kaira, Y.T., S.A., I.M., A.I., and W.W. Study supervision: S.Y.

## Competing interests

Y.T. obtained honoraria from Chugai Pharma, AstraZeneca, and Boehringer-Ingelheim. T. Yamada obtained research grants from Ono Pharma, Chugai Pharma, Pfizer Co, and Takeda Pharma. K. Kaira obtained research grants from AstraZeneca, and honoraria from Ono Pharma, Eli Lilly, Chugai Pharma, AstraZeneca, and Boehringer-Ingelheim. S.A. obtained grants and personal fees from Chugai, grants and non-financial support from F. Hoffmann-La Roche, grants and personal fees from Boehringer Ingelheim, grants and personal fees from AstraZeneca. S.Y. obtained research grants and honoraria from Chugai Pharma, AstraZeneca, and Boehringer-Ingelheim. The other authors have no conflict of interest.
