## [Peer Review File · Nature Communications]

Reviewers' Comments:

Reviewer #1:

Remarks to the Author:

This manuscript reports that transient inhibition of the RTK IGF-1R is an effective potential approach to eliminate EGFR mutant lung cancer cells that are AXL-low yet survive to seed osimertinib tumors. A mechanism for IGF-1R induction is provided, centering on FOXA1.

The manuscript is clear. The findings are interesting and could be clinically relevant. There are several issues that diminish enthusiasm, as described below.

-Which other RTKs were activated upon osimertinib treatment in the EGFR mutant cell lines?

-What is the mechanism by which osimertinib induces FOXA1? What is the pathway and factor that is responsible?

-The findings are very similar to those originally reported by in the initial manuscript describing drug tolerant persister cells in cancers under TKI treatment (PMID: 20371346). How do the authors move beyond that prior study? Are there epigenetic therapies that can combat the osimertinib tolerance, similar to those employed in that published work?

-What is the mechanism of constitutive adaptor protein binding to IGF-1R? This result seems counterintuitive based on current knowledge. Furthermore, what is the functional significance of the EGFR/IGF-1R interaction that appears to be stable? Does knockdown of EGFR impair the ability of IGF-1R to transduce signaling and promote drug tolerance?

-Is IGF-1R expression and phosphorylation and FOXA1 expression associated with worse response to osimertinib in EGFR mutant lung cancer patients?

Minor point: western blots for pEGFR levels should be shown in the main figures as a control during each drug treatment

Reviewer #2:

Remarks to the Author:

In this manuscript, Wang et al. uncover IGF-1R as a potential target in acquired resistance to osimertinib in AXL-low expressing EGFR mutated lung cancers. Previously, they reported a population of AXL-high expressing tumors with increased tolerance to osimertinib by restoring survival signals associated with AXL, EGFR, and HER3 (Nat Commun. 2019 Jan 16;10(1):259.). Here, they focus on AXL-low expressing tumor cells which are more sensitive to osimertinib than AXL-high expressing tumor cells, but exhibit persister cells with resistance to osimertinib. In vitro studies showed that levels of phosphorylated (pIGF-1R) were increased in AXL-low tumor cell lines (HCC4006, HCC827, and H3255) compared to AXL-high tumor cell lines, (HCC4011, PC9, and PC9/GXR). Furthermore, pIGF-1R increased after osimertinib exposure in AXL-low tumor cell lines but not in AXL-high tumor cell lines. They also conducted a systematic set of studies which identified FOXA1 as the transcription factor that activates IGF-1R upregulation by osimertinib exposure in HCC827 cells. Co-immunoprecipitation experiments showed that treatment with osimertinib increased association of IGF-1R with adaptor proteins, IRS1 and GAB1 but not EGFR. Finally, in vivo experiments showed combined treatment with osimertinib and linsitinib, IGF-1R inhibitor could inhibit tumor growth compared to osimertinib alone in AXL-low tumor cell lines. The therapeutic effect was confirmed even with transient combination treatment considering feasibility of linsitinib in both AXL-low tumor cell lines and PDX models. They concluded that IGF-1R is an important target in AXL-low expressing EGFR mutated lung cancers.

While the data in this study are solid, the idea of targeting IGF-1R in EGFR mutant lung cancer is not a novel concept, including with 3rd generation EGFR TKIs (eg Cortot et al., Cancer Research 2013 Jan 15; 73 (2) 834). Thus I'm not sure these findings advance the field suitably to be published in Nature Communications.

Additional Comments

1. Fig 1E – Additional RTKs appear to be activated in HCC4006 and H3255 by osimertinib (eg top row, middle dots for HCC4006 and second row, left dots for H3255). What are these RTKs? While the authors have focused on IGF1R it appears that activation may be more heterogeneous.
2. Fig 2 – Would be more convincing using multiple IGF1R CRISPR-CAS9 KO given potential off target effects of IGF1R siRNAs.
3. Fig 3 – Again would be strengthened by multiple FOXA1 CRISPR-CAS9 KO instead of using a single FOXA1 shRNA.
4. To test whether AXL directly mediates this altered dependency on IGF1R, functional experiments examining the impact of over-expressing AXL in AXL-low cells would also significantly strengthen the manuscript, especially since this is the only novel point compared with prior work.
5. Methods section is missing description of flow cytometry

Reviewers' comments:

Reviewer #1 (Remarks to the Author):

This manuscript reports that transient inhibition of the RTK IGF-1R is an effective potential approach to eliminate EGFR mutant lung cancer cells that are AXL-low yet survive to seed osimertinib tumors. A mechanism for IGF-1R induction is provided, centering on FOXA1.

The manuscript is clear. The findings are interesting and could be clinically relevant. There are several issues that diminish enthusiasm, as described below.

1-Which other RTKs were activated upon osimertinib treatment in the EGFR mutant cell lines?

Answer:

As indicated by the reviewers, phosphorylation of MET (HGF-R, point C3, C4) and FGFR3 (point B13, B14) was increased in H3255 and HCC4006 cells, respectively, following osimertinib treatment, as shown in Fig 1E. We evaluated the phosphorylation of MET and FGFR3 by western blotting. However, we detected no discernible increase in phosphorylated MET or FGFR3 in H3255, HCC4006, or HCC827 cells treated with osimertinib for 72 h (A). We further assessed the effect of a MET inhibitor (crizotinib) and FGFR inhibitor (BGJ398) on the osimertinib sensitivity of H3255 and HCC4006 cells, respectively, using the MTT assay. Although crizotinib and BGJ398 inhibited the phosphorylation of MET and FGFR3 in H3255 and HCC4006 cells, respectively, neither crizotinib nor BGJ398 remarkably affected the osimertinib sensitivity of H3255 (B, C) and HCC4006 (D, E) cells, respectively. These results indicate that MET and FGFR3 are unlikely to play predominant roles in mediating osimertinib sensitivity in *EGFR* mutated lung cancer cells tested, at least, in our experimental conditions.

These results were included in the Results section (p6, lines 11-21) and **Supplementary Figure 3**.

A

2-What is the mechanism by which osimertinib induces FOXA1? What is the pathway and factor that is responsible?

Answer:

We would like to thank the reviewer for this scientifically relevant question. We performed additional experiments and uncovered the mechanism by which FOXA1 could be upregulated following osimertinib exposure.

In contrast to the IGF-1R expression results shown in **Supplementary Fig. 4A** of the original version, FOXA1 induction following osimertinib exposure was not impacted by cycloheximide treatment, indicating that FOXA1 upregulation by osimertinib does not require de novo protein synthesis (**Figure 3G**). We hypothesized that pre-existing signaling proteins or pathways might be responsible for the induction of FOXA1 mRNA by osimertinib. Accordingly, we observed that osimertinib-dependent FOXA1 induction was significantly inhibited in IGF-1R knockout HCC827 cell clones (**Figure 3H**). These results suggested that endogenous IGF-1R protein was involved in the signal transduction activating FOXA1 mRNA expression following osimertinib exposure. These new data indicated that osimertinib exposure activated FOXA1 expression through the signaling pathway comprising endogenous IGF-1R protein. Then, FOXA1 induced the transcriptionally more active epigenetic status of the IGF-1R gene, resulting in the positive feedback activation of IGF-1R in HCC827 cells (**Figure 3J**).

These findings are now stated in the revised manuscript (p8, lines 3-32). Additionally, these new findings were included in **Figure 3**. The schema was also revised based on these findings (**Figure 3J**).

3-The findings are very similar to those originally reported by in the initial manuscript describing drug tolerant persister cells in cancers under TKI treatment (PMID: 20371346). How do the authors move beyond that prior study? Are there epigenetic therapies that can combat the osimertinib tolerance, similar to those employed in that published work?

Answer:

As highlighted by the reviewer, IGF-1R involvement in the emergence of tolerant cells to first-generation EGFR-TKIs has been previously reported. Sharma et al have shown that first-generation EGFR-TKIs stimulated the expression of KDM5A, a histone demethylase, resulting in increased IGF-1R phosphorylation, associated with IGF-BP3 overexpression, and thereby inducing drug-tolerant cells. Furthermore, they reported that the combined use of the IGF-1R inhibitor, AEW541, and gefitinib inhibited the emergence of drug-tolerant cells *in vitro*. However, these findings were mainly observed using PC-9 cells, which express a high level of AXL (please see Fig 1B of our paper), and the efficacy of the combined treatment *in vivo* has not been reported.

In the present study, we demonstrated novel findings focused on AXL-low expressing *EGFR* mutated NSCLC, which are more sensitive to EGFR-TKIs when compared with AXL-high expressing *EGFR* mutated NSCLC. In this population of cell lines, tolerance to the clinically available third-generation EGFR-TKI osimertinib is induced via increased IGF-1R protein expression. We further uncovered the mechanism by which osimertinib increased IGF-1R expression. Osimertinib exposure activated expression of transcription factor FOXA1 through the signaling pathway comprising endogenous IGF-1R protein. Then, FOXA1 induced the transcriptionally more active epigenetic status of the IGF-1R gene, resulting in the positive feedback activation of IGF-1R. Moreover, we demonstrated the efficacy of a new therapeutic strategy, the transient combination of IGF-1R inhibition with continuous osimertinib treatment, revealing remarkable improvements in the outcome of AXL-low expressing *EGFR* mutated NSCLC. We believe that these novel findings are

crucial in considering therapeutic strategies that cure or dramatically improve the prognosis of *EGFR* mutated lung cancer.

Regarding epigenetic therapy, the development of drugs that inhibit FOXA1 might be necessary. We aim to develop FOXA1 inhibitors and report their efficacy on osimertinib tolerance in future reports.

These statements were included in the Discussion section (p11 line 27-p12 line 1, p12 lines 6-9, and p12 lines 32-33).

4-What is the mechanism of constitutive adaptor protein binding to IGF-1R? This result seems counterintuitive based on current knowledge. Furthermore, what is the functional significance of the EGFR/IGF-1R interaction that appears to be stable? Does knockdown of EGFR impair the ability of IGF-1R to transduce signaling and promote drug tolerance?

Answer:

Thank you for this important question. As indicated by the reviewer, recent review articles have presented the importance of ligand binding for adaptor protein binding to IGF-1R as follows, “Ligand binding induces a conformational change that activates the kinase domain of IGF-1R resulting in autophosphorylation of specific tyrosine residues, which appears to be the critical step in receptor activation. This in turn leads to recruitment and phosphorylation of the docking proteins insulin receptor substrates (IRS-1/2) and Shc, ultimately resulting in the activation of multiple signaling pathways, of which the two most well-characterized are (PI3K-AKT) and RAS- MAPK.” (Osher E and Macaulay VM, *Cells* 2019; 8, 895) Conversely, other review articles have presented the crosstalk between IGF-1R and other receptor tyrosine kinases (RTKs), including EGFR (Liu C et al, *Curr Pharm Des* 2014;20:2912-21). Therefore, even in the steady-state, crosstalk with RTKs containing EGFR may induce a certain activation of IGF-1R and bind scaffold proteins. We performed additional experiments to examine the interaction between IGF-1R and mutated EGFR protein, using exon19 deleted EGFR specific antibody (EGFR-del19). We observed that osimertinib increased the binding of IGF-1R to total EGFR protein, as well as mutated EGFR protein. These new findings indicate that IGF-1R is associated with mutated EGFR, irrespective of osimertinib exposure (**Figure 1A for reviewer only**).

As suggested by the reviewer, we also performed additional experiments with EGFR knockdown to assess the ability of IGF-1R on promoting drug tolerance. We successfully knocked down EGFR expression in HCC4006 cells by siRNA for EGFR (**Figure 1B for reviewer only**). The knockdown of EGFR expression resulted in decreasing cell viability, and linsitinib (IGF-1R inhibitor) further inhibited the viability (**Figure 1C for reviewer only**). However, since HCC4006 cells treated with siRNA for EGFR still expressed a discernible level of EGFR, we could not directly evaluate the ability of IGF-1R in the absence of EGFR on promoting osimertinib tolerance.

5-Is IGF-1R expression and phosphorylation and FOXA1 expression associated with worse response to osimertinib in EGFR mutant lung cancer patients?

Answer:

We agree that the relevance of the target molecules needs to be demonstrated using clinical specimens. However, in this study, it is currently challenging to present the relevance of IGF-1R and FOXA1 expression in clinical specimens. Our *in vitro* data indicated that the expression of IGF-1R and FOXA1 was upregulated after osimertinib exposure (**Figure 1F and Supplementary Table 1**). For example, the level of constitutively expressed IGF-1R did not correlate with osimertinib sensitivity in *EGFR* mutated lung cancer cell lines (**Figure 1B**). We collected clinical specimens obtained before osimertinib treatment (**Figure 1C**). To demonstrate the induction of IGF-1R and FOXA1, clinical specimens obtained a few days after osimertinib treatment initiation are required, but could be impossible in clinical practice.

Conversely, our cell line data indicated that AXL-low expressing, osimertinib highly sensitive *EGFR* mutated NSCLC cell lines constitutively expressed high levels of phosphorylated IGF-1R (pIGF-1R)(**Figure 1B**). Therefore, we immunohistochemically evaluated pIGF-1R expression in the pre-EGFR-TKI-treated tumor samples for pIGF-1R using an anti-pIGF-1R antibody (Cell Signaling, p-IGF-1R (Tyr1131/IR (Tyr1146)) used in previous reports (Kruger DT et al, Int J Cancer 146;2348-2359; Björner S et al, Oncotarget 8;9093-9107). Our preliminary findings revealed that the subcutaneous tumor produced by HCC827 cells (constitutive level of pIGF-1R expression *in vitro* is

high) was strongly stained, but the tumor produced by PC-9 cells (constitutive level of pIGF-1R expression *in vitro* is low) was not, indicating the specificity of this antibody in IHC (**Figure 2 for reviewer only**).

As additional clinical specimens obtained from 16 patients demonstrated low levels of AXL expression in all tumor cells, we stained these specimens with the anti-IGF-1R antibody, with scoring (Supplementary Figure 3) performed similarly to AXL staining (very high (3+), high (2+), low (1+), and no (0) expression) (**Supplementary Figure 4A**). Of the 16 specimens, high (2+ to 3+) and low (no to 1+) pIGF-1R expression was observed in 5 (31%) and 11 (69%) specimens, respectively. In patients with pIGF-1R-high expression, the response rate to osimertinib was 100% (5/5); however, in patients with pIGF-1R-low expression, the response rate was 73% (8/11) (**Supplementary Figure 4B**). Additionally, in patients with pIGF-1R-high expression, osimertinib induced tumor shrinkage more remarkably when compared with pIGF-1R-low expression ($p=0.011$) (**Supplementary Figure 4C**).

We stated these findings in the Results section (p6, lines 22-31).

A. IHC for pIGF-1R

B. Waterfall plot

C. pIGF-1R expression and tumor shrinkage

6 Minor point: western blots for pEGFR levels should be shown in the main figures as a control during each drug treatment

Answer:

As indicated by the reviewer, data for phosphorylated EGFR and total EGFR were missing in **Figure 3B and D**. Hence, we added the data for total EGFR and phosphorylated EGFR. **Figure 3D**, in the original version, is now listed as **Figure 3E** because, as requested by Reviewer #2, we inserted new data as **Figure 3D** in the revised version.

Reviewer #2 (Remarks to the Author):

In this manuscript, Wang et al. uncover IGF-1R as a potential target in acquired resistance to osimertinib in AXL-low expressing EGFR mutated lung cancers. Previously, they reported a population of AXL-high expressing tumors with increased tolerance to osimertinib by restoring survival signals associated with AXL, EGFR, and HER3 (Nat Commun. 2019 Jan 16;10(1):259.). Here, they focus on AXL-low expressing tumor cells which are more sensitive to osimertinib than AXL-high expressing tumor cells, but exhibit persister cells with resistance to osimertinib. In vitro studies showed that levels of phosphorylated (pIGF-1R) were increased in AXL-low tumor cell lines (HCC4006, HCC827, and H3255) compared to AXL-high tumor cell lines, (HCC4011, PC9, and PC9/GXR). Furthermore, pIGF-1R increased after osimertinib exposure in AXL-low tumor cell lines but not in AXL-high tumor cell lines. They also conducted a systematic set of studies which identified FOXA1 as the transcription factor that activates IGF-1R upregulation by osimertinib exposure in HCC827 cells. Co-immunoprecipitation experiments showed that treatment with osimertinib increased association of IGF-1R with adaptor proteins, IRS1 and GAB1 but not EGFR. Finally, in vivo experiments showed combined treatment with osimertinib and linsitinib, IGF-1R inhibitor could inhibit tumor growth compared to osimertinib alone in AXL-low tumor cell lines. The therapeutic effect was confirmed even with transient combination treatment considering feasibility of linsitinib in both AXL-low tumor cell lines and PDX models. They concluded that IGF-1R is an important target in AXL-low expressing EGFR mutated lung cancers.

While the data in this study are solid, the idea of targeting IGF-1R in EGFR mutant lung cancer is not a novel concept, including with 3rd generation EGFR TKIs (eg Cortot et al., Cancer Research 2013 Jan 15; 73 (2) 834). Thus I'm not sure these findings advance the field suitably to be published in Nature Communications.

Answer:

Cortot et al have reported that IGF-1R activation as a result of the loss of IGFBP3 expression, via methylation of its promoter lesions, is involved in the emergence of tolerant or resistant cells to EGFR-TKIs, PF299804 and WZ4002. Furthermore, they have reported that the combined use of IGF-1R inhibitors, BMS536924 and OSI-906 (Linsitinib), restored sensitivity to EGFR-TKIs *in vitro*. These findings were observed in resistant cells obtained from PC-9 cells, expressing a high level of AXL (please see **Figure 1B** in our paper). Moreover, the efficacy of combined treatment with EGFR-TKI and IGF-1R inhibitors *in vivo* has not been reported.

In the present study, we revealed novel findings focusing on AXL-low expressing *EGFR* mutated NSCLC, which are more sensitive to EGFR-TKIs when compared with AXL-high expressing *EGFR* mutated NSCLC. In this population of cell lines, tolerance to the clinically available third-generation EGFR-TKI osimertinib is induced via increased IGF-1R protein expression, distinct from mechanisms reported by Cortot et al (Cancer Res 2013) and Sharma et al (Cell 2010). We further uncovered the mechanism by which osimertinib increased IGF-1R expression. Osimertinib exposure activated the expression of transcription factor FOXA1 through the signaling pathway comprising the endogenous IGF-1R protein. Then, FOXA1 induced the transcriptionally more active epigenetic status of the IGF-1R gene, resulting in the positive feedback activation of IGF-1R. Moreover, we demonstrated the efficacy of a new therapeutic strategy, the transient combination of IGF-1R inhibition with continuous osimertinib treatment, showing remarkable improvement in the outcome of AXL-low expressing *EGFR* mutated NSCLC. We believe that these novel findings are crucial in considering therapeutic strategies that cure or dramatically improve the prognosis of EGFR mutated lung cancer.

We added these statements in the Discussion section (p11 line 27-p12 line 1, p12 lines 6-9, and p12 lines 32-33).

Additional Comments

2. Fig 2 – Would be more convincing using multiple IGF1R CRISPR-CAS9 KOs given potential off target effects of IGF1R siRNAs.

Answer:

As suggested by the reviewer, we established IGF-1R knockout clones for HCC827 cells using CRISPR-CAS9 for two different sites of IGF-1R (KO1 and KO2) (**Supplementary Figure 6**). Finally, we obtained two clones (KO1.6 and KO1.21) using KO1 and one clone with KO2 (KO2.14). These three clones were more sensitive to osimertinib when compared with parental HCC827 cells. As these clones were markedly sensitive to osimertinib and protein was harvested from viable cells

after 72 h osimertinib exposure, signal transduction in cells was examined after 2 h osimertinib exposure by western blotting. In the IGF-1R knockout clone (KO1-6), osimertinib inhibited the phosphorylation of GAB1, Shc, AKT, and ERK more remarkably when compared with HCC827 cells.

We added the data and statements for IGF-1R CRISPR-CAS9 in **Figure 2 D, E, and F.** and in the Results section (p7, lines 12-17), respectively. We also added the statements for methods of “generation of IGF-1R knockout HCC827 cells by CRISPR/Cas9 system” in the Methods section (p15, line 26 – p16, line 9). We added the reference for CRISPR/Cas9 system as reference 42. Accordingly, we renumbered references 43 and 44 in the revised version of manuscript. Since we obtained CRISPR/Cas9 plasmid from Dr Ryo Imamura, we stated his name in the Acknowledgements.

3. Fig 3 – Again would be strengthened by multiple FOXA1 CRISPR-CAS9 KOs instead of using a single FOXA1 shRNA.

Answer:

We agree with the reviewers’ comments. The Associate Editor kindly mentioned that “if this is not possible please include additional shRNAs for validation.” Therefore, we examined the effect of two additional shRNAs. Overall, three different shRNA for FOXA1 consistently suppressed the increased IGF-1R expression and its phosphorylation. Additionally, they inhibited colony formation of HCC827 cells in the presence of osimertinib. We added the data regarding the additional two FOXA1-shRNA in **Figure 3 C and F.**

4. To test whether AXL directly mediates this altered dependency on IGF1R, functional experiments examining the impact of over-expressing AXL in AXL-low cells would also significantly strengthen the manuscript, especially since this is the only novel point compared with prior work.

Answer:

As suggested, we attempted to overexpress AXL in AXL-low tumor cells. Although we obtained AXL overexpressing HCC827 cells, as well as AXL overexpressing PC-9 cells as a control, we were unable to obtain AXL overexpressing H3255 or HCC4006 cells, even on attempting overexpression using two different expression vectors (pIRESpuro and pEZ-Lv105). The transfection of AXL, but not the empty vector, was extremely toxic to H3255 cells. In HCC4006 cells, although tumor cells survived after transfection, the tumor cells failed to overexpress AXL. Therefore, we evaluated osimertinib sensitivity using HCC827 cells, as well as PC-9 cells as a control, transfected with or without AXL overexpression (**Figure 3A for reviewer only**). In PC-9 cells, AXL overexpression discernibly increased sensitivity to osimertinib (**Figure 3B for reviewer only**). In HCC827 cells, although AXL overexpression associated with AXL phosphorylation was successfully induced, it failed to explicitly affect osimertinib sensitivity (**Figure 3C for reviewer only**). Interestingly, AXL overexpression reproducibly inhibited IGF-1R protein expression in HCC827 cells (**Figure 3A for reviewer only**). Therefore, in HCC827 cells, AXL overexpression might suppress its effect on osimertinib sensitivity by decreasing IGF-1R protein expression, which may be induced by an unknown mechanism and potentially sensitize cells to osimertinib. These data are interesting, and the precise mechanisms should be intensively analyzed; however, we believe that this is beyond the scope of this paper. We would like to further analyze the mechanisms and report these results in future reports.

5. Methods section is missing description of flow cytometry

Answer:

As suggested, we added the methods for flow cytometry in the Supplementary Materials and Methods section (p15, lines 8-18).

Reviewers' Comments:

Reviewer #1:

Remarks to the Author:

In the revised manuscript, the authors have addressed the comments of the reviewers. In general the preclinical aspects of the manuscript are improved. However, the new clinical specimen data were added in response to one of my comments seem paradoxical with the authors main conclusion. High pIGF1R expression in tumor biopsies from EGFR mutant patients treated with an EGFR TKI was associated with improved response to EGFR TKI treatment. If this is a mechanism of drug tolerance/resistance as the authors claim, then one would expect the opposite result.

Reviewer #2:

Remarks to the Author:

The authors have satisfactorily addressed my concerns.

Response to comments

Reviewer #1 (Remarks to the Author):

In the revised manuscript, the authors have addressed the comments of the reviewers. In general the preclinical aspects of the manuscript are improved. However, the new clinical specimen data were added in response to one of my comments seem paradoxical with the authors main conclusion. High pIGF1R expression in tumor biopsies from EGFR mutant patients treated with an EGFR TKI was associated with improved response to EGFR TKI treatment. If this is a mechanism of drug tolerance/resistance as the authors claim, then one would expect the opposite result.

Answer:

Thank you for this careful comment. As indicated by the reviewer, in experiments performed using clinical specimens, we showed that tumors with high levels of constitutive expression of phosphorylated IGF-1R were more sensitive to osimertinib than those with low levels of constitutive expression of phosphorylated IGF-1R (**Supplementary Figure 4B, 4C**), though the number of specimens was limited. In addition, via *in vitro* experiments, which were conducted using *EGFR*-mutated NSCLC cell lines, we demonstrated that tumor cells with high constitutive expression of phosphorylated IGF-1R were more sensitive to osimertinib than those with low expression of phosphorylated IGF-1R (**Figure 1A, 1B**). Therefore, the relationship between high sensitivity to osimertinib and high level of constitutive expression of phosphorylated IGF-1R was consistent and there was no contradiction in the results. However, the mechanisms underlying this phenomenon remain to be elucidated in the future.

Moreover, we clarified that the expression of phosphorylated IGF-1R is further increased upon exposure to osimertinib through the FOXA1/IGF-1R axis and leads to osimertinib tolerance. Based on these results and the findings of experiments conducted using cell line-derived- and patient-derived-xenograft models, we concluded that transient IGF-1R inhibition combined with osimertinib might prevent osimertinib tolerance. Therefore, the results of this study are consistent.

To make these results clearer, we have revised the statements in the Results section (p6, lines 30-36) as follows:

“Additionally, osimertinib induced tumor shrinkage more markedly in patients with pIGF-1R-high expression when compared with pIGF-1R-low expression ($p=0.011$) (**Supplementary Figure 4C**), consistent with the results of the experiments performed using *EGFR* mutated NSCLC cell lines (**Figure 1A, B**). These results suggested that while the levels of phosphorylated IGF-1R correlated with better response to osimertinib, further increase in the phosphorylation of IGF-1R in association with osimertinib-mediated increase in the protein expression of IGF-1R may play a pivotal role in the tolerance observed in AXL-low expressing *EGFR* mutated NSCLCs.”

Reviewers' Comments:

Reviewer #1:

Remarks to the Author:

The authors have addressed my comments. That baseline pIGF-1R correlates with improved osimertinib response is not logical if pIGF-1R is a mechanism of osimertinib tolerance. At the least, the authors need to comment on this in the manuscript discussion so the gap and lack of explanation are clear.

Reviewer #1 (Remarks to the Author):

The authors have addressed my comments. That baseline pIGF-1R correlates with improved osimertinib response is not logical if pIGF-1R is a mechanisms of osimertinib tolerance. At the least, the authors need to comment on this in the manuscript discussion so the gap and lack of explanation are clear.

Answer:

We agree that we should fill the gap with an explanation on the lack of information on the indicated issue. Therefore, we have added the following statements in the Discussion section (p11, line 26 - p12, line 1).

“Interestingly, the AXL-low-expressing tumor cells, which were more sensitive to osimertinib had a higher baseline level of phosphorylated IGF-1R than the AXL-high-expressing tumor cells, suggesting that phosphorylated IGF-1R at baseline level might have less impact on osimertinib sensitivity than AXL expression. Nevertheless, in the AXL-low-expressing tumor cells, through increased protein expression and phosphorylation of IGF-1R by osimertinib exposure (e.g., for 3 days), tolerance emerged via epigenetic modification. This increased expression of the transcription factor FOXA1 and restored the survival signal via IGF-1R associated with Gab1 and IRS1. The reasons why Gab1/IRS1 signaling is caused by increased IGF-1R phosphorylation via FOXA1 by osimertinib exposure but not baseline IGF-1R phosphorylation remains to be elucidated. Furthermore, the reason why baseline level expression of phosphorylated IGF-1R is higher in the AXL-low-expressing tumor cells than that in the AXL-high-expressing tumor cells is also unclear at present. Further examinations are warranted to clarify these mechanisms in the future.”